# The structure-selective endonucleases GEN1 and MUS81 mediate complementary functions in safeguarding the genome of proliferating B lymphocytes

Keith Conrad Fernandez[1,2†], Laura Feeney[3†], Ryan M Smolkin[1,4], Wei-Feng Yen[1,5], Allysia J Matthews[1,2], William Alread[1], John HJ Petrini[3,4,5]*, Jayanta Chaudhuri[1,2,4]*

[1]Immunology Program, Memorial Sloan Kettering Cancer Center, New York, United States; [2]Immunology and Microbial Pathogenesis Program, Weill Cornell Graduate School of Medical Sciences, Cornell University, New York, United States; [3]Molecular Biology Program, Memorial Sloan-Kettering Cancer Center, New York, United States; [4]Gerstner Sloan Kettering Graduate School of Biomedical Sciences, New York, United States; [5]Biochemistry, Cellular and Molecular Biology Allied Program, Weill Cornell Graduate School of Medical Sciences, Cornell University, New York, United States

*For correspondence:
petrinij@mskcc.org (JHJP);
chaudhuj@mskcc.org (JC)

†These authors contributed
equally to this work

Competing interest: The authors
declare that no competing
interests exist.

Reviewing Editor: Wolf-Dietrich
Heyer, University of California,
Davis, United States

**Abstract** During the development of humoral immunity, activated B lymphocytes undergo vigorous proliferative, transcriptional, metabolic, and DNA remodeling activities; hence, their genomes are constantly exposed to an onslaught of genotoxic agents and processes. Branched DNA intermediates generated during replication and recombinational repair pose genomic threats if left unresolved, and so they must be eliminated by structure-selective endonucleases to preserve the integrity of these DNA transactions for the faithful duplication and propagation of genetic information. To investigate the role of two such enzymes, GEN1 and MUS81, in B cell biology, we established B-cell conditional knockout mouse models and found that deletion of GEN1 and MUS81 in early B-cell precursors abrogates the development and maturation of B-lineage cells while the loss of these enzymes in mature B cells inhibits the generation of robust germinal centers. Upon activation, these double-null mature B lymphocytes fail to proliferate and survive while exhibiting transcriptional signatures of p53 signaling, apoptosis, and type I interferon response. Metaphase spreads of these endonuclease-deficient cells show severe and diverse chromosomal abnormalities, including a preponderance of chromosome breaks, consistent with a defect in resolving recombination intermediates. These observations underscore the pivotal roles of GEN1 and MUS81 in safeguarding the genome to ensure the proper development and proliferation of B lymphocytes.

## Editor's evaluation

This manuscript is of interest to individuals working on genome stability and B lymphocyte development. Using knockouts for the genes encoding the structure-selective endonucleases GEN1 and MUS81 in mice, the authors show that the absence of both proteins is incompatible with embryonic development. On the background of a GEN1 knockout, a MUS81 flox allele was used to study the effect on B-cell development using the Mb1-Cre and Cd23-Cre drivers, showing that the absence of both proteins leads to development and maturation defects. Selective loss in mature B cells inhibited germinal center formation. This is the first study of these enzymes in an organismic context and

in primary cells, revealing insight into the in vivo consequences of loss of GEN1 and MUS81 functions not previously accessible through studies in cultured cells.

## Introduction

B lymphocytes comprise the humoral arm of the adaptive immune system. They undergo a well-orchestrated series of clonal expansion and differentiation programs in the bone marrow (BM) to become mature B cells that reside in secondary lymphoid organs such as the spleen and lymph nodes (*LeBien and Tedder, 2008*; *Pieper et al., 2013*). During an adaptive immune response, B cells are recruited into the germinal center (GC) where they adopt one of two cellular fates before exiting the GC: memory B cells that confer immunological memory and plasma cells that produce antibodies of high affinity and specificity (*Mesin et al., 2016*; *Victora and Nussenzweig, 2022*). B lymphocytes are unique among other immune cells in that they initiate programmed double-strand breaks (DSBs) both as developing precursors in the BM and as GC B cells in the secondary lymphoid organs (*Alt et al., 2013*). V(D)J recombination generates a primary repertoire of B cell receptors that can be further diversified when GC B cells employ the DNA-modifying enzyme activation-induced cytidine deaminase (AID) to instigate the formation of DSBs in the immunoglobulin heavy loci (IgH) for class-switch recombination and to somatically hypermutate the variable regions of the immunoglobulin loci (*Feng et al., 2020*; *Schatz and Swanson, 2011*; *Xu et al., 2012*). To outcompete other clonal cells and be selected for survival and differentiation, GC B cells not only are required to express the correct antigen-specific receptor of high affinity and specificity, but they must also satisfy the formidable replicative, transcriptional, and metabolic demands for clonal expansion (*Young and Brink, 2021*). These cellular activities pose significant collateral genotoxic hazards to GC B cells; therefore, safeguarding the cells' genomic integrity is paramount for accurate duplication and propagation of genetic information.

Impediments to the progression of the replication fork—termed replication stress—represent a significant endogenous source of DSBs in proliferating cells, producing up to 50 DSBs per cell cycle in a cell (*Mehta and Haber, 2014*; *Zeman and Cimprich, 2014*). Factors that impair the functionality of the replication machinery include repetitive sequences, secondary structures (such as R-loops and G-quadruplexes), transcription-replication conflicts, lesions such as thymidine dimers and single-stranded breaks, oxidative stress, and imbalance or depletion of the nucleotide pool (*Zeman and Cimprich, 2014*). To ensure completion of DNA synthesis before mitosis commences, replication forks that have stalled or collapsed can be restarted via recombination-dependent or -independent pathways, the choice of which is contingent upon the nature of the replication barrier, the duration of stalling, the nature of the intermediates generated after fork stalling, and the type of processing these intermediates undergo (*Berti et al., 2020*; *Petermann and Helleday, 2010*; *Zeman and Cimprich, 2014*). Though primarily studied in the context of DSB repair, proteins involved in homologous recombination (HR) including BRCA2 and RAD51 are also critical for the protection, remodeling, and recombination-dependent restart of replication forks, highlighting the necessity of HR in mitigating replication stress and assisting the timely completion of DNA synthesis (*Ait Saada et al., 2018*; *Carr and Lambert, 2013*; *Scully et al., 2021*).

Recombination-dependent repair of DSBs and restart of replication forks entail strand invasion and homology search of an intact duplex DNA, generating a nascent joint intermediate termed the displacement loop (D-loop) that culminates in the formation of Holliday junctions (HJs) that physically link the two sister chromatids (*Falquet and Rass, 2019*). Several mechanisms have evolved to process these intermediates, as failure to eliminate them prohibits chromosomal segregation during mitosis, interfering with the faithful transmission of genetic material to the daughter cells (*West and Chan, 2017*). The BLM-TOP3A-RMI1-RMI2 (BTR) complex dissolves double HJs to generate non-crossover products while structure-selective endonucleases (SSEs) such as the SLX1-SLX4, MUS81-EME1, and XPF-ERCC1 (SMX) trinuclease complex and GEN1 resolve single and double HJs to generate both crossover and non-crossover products, depending on the position of the nicks introduced (*Blanco and Matos, 2015*). Due to their in vitro 3'-flap endonuclease activity and the formation of 3' flaps during synthesis-dependent strand annealing (SDSA), MUS81 complexes may also participate in the resolution of non-HJ-mediated recombination by cleaving such overhangs (*Hollingsworth and Brill, 2004*). Moreover, *mus81* and *yen1* in *Saccharomyces cerevisiae* can process D-loops to influence the pathway

choice and outcome of HR-mediated DSB repair as the absence of these enzymes promote break-induced replication (BIR) instead of SDSA (*Ho et al., 2010*). Because these SSEs are active against a broad spectrum of branched DNA structures, they are subjected to multiple cell cycle-dependent regulatory mechanisms so that replication can proceed without interference and that toxic recombination outcomes due to uncontrolled cleavage are minimized (*Wild and Matos, 2016*). Although the deletion of BLM itself causes embryonic lethality in mice, individual loss of GEN1 or MUS81 does not confer a strong DNA repair-deficient phenotype in unperturbed cells, implying some degree of functional overlap between the two proteins (*McDaniel et al., 2003*; *Sarbajna et al., 2014*). Only when GEN1 and MUS81 are both absent is the genomic integrity of the cells severely subverted, resulting in aberrant mitotic structures including bulky chromatin bridges and ultrafine anaphase bridges (UFBs) that lead to compromised viability, gross chromosomal mis-segregation and abnormalities, multinucleation, and heightened formation of micronuclei (*Chan et al., 2018*; *Chan and West, 2018*; *Garner et al., 2013*; *Sarbajna et al., 2014*; *Sarlós et al., 2017*; *Wechsler et al., 2011*; *West et al., 2015*).

Besides removing HR intermediates arising from recombinational DNA transactions, both SSEs process persistent replication intermediates to promote the completion of genome replication and sister chromatid disentanglement (*Falquet and Rass, 2019*). MUS81 complexes can cleave replication forks, structures resembling intact HJs such as four-way reversed forks, and D-loops to initiate and regulate BIR as a mechanism to repair dysfunctional forks (*Hanada et al., 2007*; *Hua et al., 2022*; *Kikuchi et al., 2013*; *Mayle et al., 2015*; *Pepe and West, 2014a*; *Pepe and West, 2014b*). MUS81 is also essential for the 'expression' of common fragile sites (CFSs)—sites that are prone to under-replication during stressed conditions—by cleaving stalled replication forks to enable POLD3-mediated mitotic DNA synthesis (MiDAS) (*Debatisse et al., 2012*; *Minocherhomji et al., 2015*; *Naim et al., 2013*; *Ying et al., 2013*). Generation of DSBs via MUS81-dependent cleavage of stalled replication forks, however, is not a prerequisite for HR-dependent replication restart as the uncoupling of the DNA strands at collapsed forks generates ssDNA that can invade and re-initiate DNA synthesis (*Lambert et al., 2010*; *Rass, 2013*). Though GEN1 has been implicated in maintaining replication fork progression alongside MUS81 and in eliminating persistent replication intermediates in Dna2 helicase-defective yeast cells to potentially facilitate MiDAS, its importance in rectifying replication stress and DNA under-replication in a genetically intact background remains to be determined (*Ölmezer et al., 2016*; *Sarbajna et al., 2014*).

The role of GEN1 and MUS81 in replication-challenged in vitro settings employing ionizing irradiation, DNA-damaging chemicals, and replication inhibitors is well established. Less is known, however, of these enzymes' importance in an unperturbed in vivo context as the use of such genotoxic agents may not accurately reproduce the nature and composition of various DNA lesions generated during cellular proliferation. Because B cells face an elevated risk of cell death and oncogenic transformation due to the high level of replication stress and DSBs inflicted on their genome by AID-dependent and -independent activities (*Alt et al., 2013*; *Barlow et al., 2013*; *Basso and Dalla-Favera, 2015*; *Macheret and Halazonetis, 2015*), we asked whether B cells require GEN1 and MUS81 to develop, survive, and perform their immunological functions. By employing different stage-specific Cre strains and a global *Gen1*-knockout mouse carrying floxed *Mus81* alleles, we report that the loss of both *Gen1* and *Mus81* in early pro-B cells severely impaired B cell development whereas double-null mature B cells failed to form competent GCs at steady-state and after immunization. Ex vivo characterization of the endonuclease-knockout cells uncovered a proliferation block caused by G2/M arrest, potent activation of p53 and apoptotic pathways, induction of type I interferon (IFN) response, and widespread chromosomal aberrations. Our findings support the notion that the function of GEN1 and MUS81 in highly proliferative somatic cells such as B cells is in eliminating replication- and recombination-born junction and branched DNA molecules ranging from stalled and reversed replication forks to D-loops and dHJs to maintain replication and genomic fidelity, ensure proper chromosome segregation, and avert mitotic catastrophe.

## Results
### GEN1 and MUS81 are critical to normal B cell lymphopoiesis
We analyzed a publicly accessible RNA-seq data set (GSE72018) to ascertain the expression pattern of *Gen1* and *Mus81* in various developing and mature B cell subsets (*Brazão et al., 2016*). Expression

of *Gen1* is higher in the proliferating pro-B, pre-B, and GC B cells than in follicular B, marginal zone B, and peritoneal B1a cells (**Figure 1A**). Conversely, *Mus81* RNA expression is relatively similar across all B cell subsets examined except in peritoneal B1a cells (**Figure 1B**). RT-qPCR analysis of primary splenic naïve B lymphocytes stimulated in culture with lipopolysaccharide (LPS) and interleukin-4 (IL-4) revealed a 10-fold increase in *Gen1* expression as early as 24-hr post-activation, while the expression of *Mus81* was not altered following activation (**Figure 1—figure supplement 1A**). We surmise that *Gen1* expression is more closely associated with the activation and proliferation of B cells than is *Mus81* expression.

To interrogate the roles of GEN1 and MUS81 in B cell development and function, we generated *Gen1*$^{-/-}$ mice by deleting the XPG nuclease domain encoded in exon 4 of *Gen1* and subsequently crossed them to *Mus81*$^{-/-}$ mice (**Dendouga et al., 2005**; **Figure 1—figure supplement 1B**). Whereas *Gen1*$^{-/-}$ and *Mus81*$^{-/-}$ mice were born at the expected Mendelian frequencies, no live *Gen1*$^{-/-}$ *Mus81*$^{-/-}$ pups were produced, indicating that constitutive loss of both GEN1 and MUS81 leads to embryonic death (**Figure 1—figure supplement 1C**). To circumvent this lethality, we generated floxed *Mus81* (*Mus81*$^{fl/fl}$) mice in which exons 3–10 encoding for the ERCC4 nuclease domain are flanked by loxP sites (**Figure 1—figure supplement 1D**). We bred these mice to the *Mb1* (*Cd79a*)-Cre strain, wherein Cre recombinase expression is driven by the *Cd79a* promoter and the excision of the floxed allele is initiated at the pre-pro B cell stage (**Fahl et al., 2009**; **Hobeika et al., 2006**). The *Mb1*-Cre: *Mus81*$^{fl/fl}$ mice were mated to *Gen1*$^{-/-}$ *Mus81*$^{fl/fl}$ mice to produce the conditional knockout *Mb1*-Cre: *Gen1*$^{-/-}$ *Mus81*$^{fl/fl}$, the single knockouts *Gen1*$^{-/-}$ *Mus81*$^{fl/fl}$ and *Mb1*-Cre: *Gen1*$^{+/-}$ *Mus81*$^{fl/fl}$, and the control *Gen1*$^{+/-}$ *Mus81*$^{fl/fl}$.

We analyzed the spleen of the *Mb1*-Cre *Gen1*$^{-/-}$ *Mus81*$^{fl/fl}$ mice and found that its cellularity was reduced by ninefold compared with that of control and single knockout mice (**Figure 1C-E**). The near complete ablation of mature B220+CD43+ and B220+CD43− cellular compartments indicated a significantly perturbed development or maintenance of peripheral B-lineage cells (**Figure 1E**). Examination of the BM revealed a 75% reduction in total cell number that was caused by the severe loss of the total B cell population; the T cell fraction remained unaltered (**Figure 1F and G**). Quantification of the B-lineage subpopulations showed that both immature and mature recirculating B cells were absent in the *Mb1*-Cre *Gen1*$^{-/-}$ *Mus81*$^{fl/fl}$ BMs (**Figure 1H**), and that the endonuclease-deficient B cell progenitors produced few CD19$^+$ CD43$^+$ CD249$^-$ CD24$^{var}$ pro-B cells (**Figure 1I**). Quantitative genomic PCR of the unrecombined *Mus81*$^{fl/fl}$ allele in *Mb1*-Cre: *Gen1*$^{+/-}$ *Mus81*$^{fl/fl}$ pro-B (fractions B+C) cells confirmed efficient Cre-mediated deletion of the loxP-flanked exons (**Figure 1—figure supplement 1E**). We further validated by RT-qPCR the absence of *Gen1* and *Mus81* mRNA expression in *Gen1*$^{-/-}$ *Mus81*$^{fl/fl}$ and *Mb1*-Cre: *Gen1*$^{+/-}$ *Mus81*$^{fl/fl}$ pro-B cells, respectively (**Figure 1—figure supplement 1F**). These findings indicate that *Gen1* and *Mus81* in early B-cell precursors are necessary for the differentiation, expansion, or maintenance of developing pro-B cells.

## GEN1 and MUS81 are required for robust GC responses

To circumvent the B cell developmental block in *Mb1*-Cre: *Gen1*$^{-/-}$ *Mus81*$^{fl/fl}$ mice, we employed the *Cd23*-Cre deleter that expresses Cre specifically in naïve B cells under the control of a transgenic *Cd23* promoter (**Kwon et al., 2008**). We crossed the *Cd23*-Cre strain to *Gen1*$^{-/-}$ *Mus81*$^{fl/fl}$ mice, generating *Cd23*-Cre: *Gen1*$^{-/-}$ *Mus81*$^{fl/fl}$ (designated henceforth as DKO), *Cd23*-Cre: *Gen1*$^{+/-}$ *Mus81*$^{fl/fl}$ (*Mus81*-KO), *Gen1*$^{-/-}$ *Mus81*$^{fl/fl}$ (*Gen1*-KO), *Cd23*-Cre: *Gen1*$^{+/-}$ *Mus81*$^{fl/+}$ (Cre control), and *Gen1*$^{+/-}$ *Mus81*$^{fl/fl}$ (control) littermates. Though the B cell precursor subsets in the BM of the DKO mice did not exhibit any major alterations in their frequencies or absolute numbers (**Figure 2—figure supplement 1A-D**), the total BM cellularity was reduced by 30%, attributed to the 60% decrease in the number of the mature recirculating B cells (**Figure 2—figure supplement 1C**). In the spleens of DKO mice, the frequencies and absolute numbers of the various B cell subsets were comparable to those of control and the *Gen1*-KO and *Mus81*-KO (collectively referred to as SKO) mice (**Figure 2—figure supplement 1E-J**). RT-qPCR analysis of splenic mature DKO B cells activated with LPS+IL-4 confirmed the ablation of *Gen1* and *Mus81* transcripts (**Figure 2—figure supplement 1K**). These data show that the deletion of *Gen1* and *Mus81* in naïve, mature B cells does not markedly impact the development and maintenance of homeostatic B cell compartments; thus, the DKO mouse can serve as a genetic tool to investigate the requirement of *Gen1* and *Mus81* in activated, mature B cells.

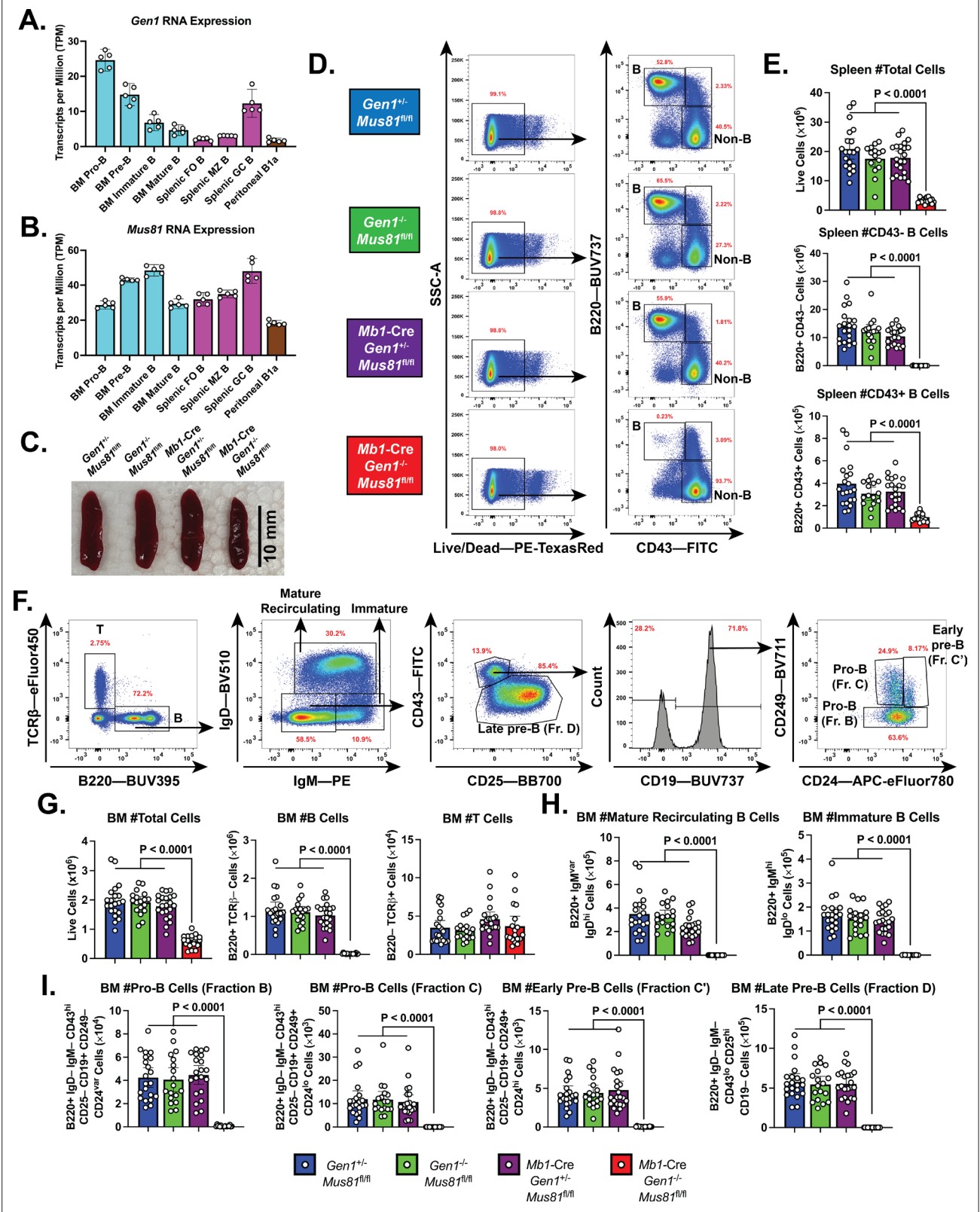

**Figure 1.** B cell development in the bone marrow (BM) and spleen of *Mb1*-Cre *Gen1*−/− *Mus81*fl/fl mice. (**A, B**) mRNA expression of *Gen1* (**A**) and *Mus81* (**B**) in developing and mature B cell subsets in the BM, spleen, and peritoneal cavity. FO: follicular, MZ: marginal zone (GEO Accession: GSE72018). (**C**) Spleens harvested from 4-month-old mice of the indicated genotypes. (**D, E**) Gating strategy (**D**) and absolute quantification (**E**) of live splenocytes, splenic B220+CD43–B, and B220+CD43+B cells. (**F**) Gating strategy of total B (B220+TCRβ–), total T (B220–TCRβ+), mature recirculating, immature,

*Figure 1 continued*

pro-B (fractions B and C), early pre-B (fraction C'), and late pre-B cells (fraction D). (**G–I**) Absolute quantification of BM cellularity, total B, and total T cell populations (**G**), of mature recirculating and immature B cell populations (**H**), and of B cell populations belonging to fractions B to D (**I**). Data in (**E**) and (**G–I**) are from four independent experiments with 18–22 mice per genotype. Bars display the arithmetic mean and error bars represent the 95% confidence interval of the measured parameters. P values were enumerated using ordinary one-way ANOVA analysis with Dunnett's multiple comparisons test without pairing wherein all means were compared to the *Mb1*-Cre *Gen1*$^{-/-}$ *Mus81*$^{fl/fl}$ group. TPM, transcripts per million.

The online version of this article includes the following figure supplement(s) for figure 1:

**Figure supplement 1.** Temporal expression of *Gen1* and *Mus81*, genetic structure of *Gen1*$^{-/-}$ and *Mus81*$^{fl/fl}$ alleles, Mendelian frequencies of *Gen1*$^{-/-}$ and *Mus81*$^{-/-}$ offsprings, and genotype validation of *Gen1*$^{-/-}$ *Mus81*$^{fl/fl}$ and *Mb1*-Cre *Gen1*$^{+/-}$ *Mus81*$^{fl/fl}$ pro-B cells.

We next characterized the impact of *Gen1* and *Mus81* deletion on the steady-state GC response in the mesenteric lymph nodes and Peyer's patches—sites where B cells continuously encounter and are activated by microbial and food antigens (*Figure 2A*). We found that the frequency and absolute number of GC B cells were decreased by 1.5- to 2-fold in DKO mice compared with control and SKO mice (*Figure 2B and C*). The presence of residual DKO GC B cells could be attributed to the selection of GC B cells that have escaped Cre-mediated deletion of the *Mus81*$^{fl/fl}$ allele, as evidenced by the detection of unrecombined *Mus81*$^{fl/fl}$ allelic copies and *Mus81* mRNA transcript within the DKO GC B cell population compared with the near-complete deletion of *Mus81*$^{fl/fl}$ allele in the non-GC B cell counterpart (*Figure 2—figure supplement 2A and B*). *Gen1* mRNA expression, however, was absent in both DKO GC and non-GC B cells (*Figure 2—figure supplement 2C*).

To assess the importance of GEN1 and MUS81 in supporting the formation and integrity of induced GCs, we immunized mice with sheep red blood cells (SRBCs) to elicit a T cell-dependent GC response. Mice were boosted 10 days after the primary dose and the GCs in the spleen were analyzed at day 14 (*Figure 2D*). Only in the DKO mice was GC formation abrogated: the frequency and absolute number of the GCs were 5% of that observed in control and SKO mice (*Figure 2E*). Within the sparse population of DKO GC B cells that we could reliably detect, the frequency of IgG1-switched cells was reduced by twofold compared to that in control and SKO GCs; more importantly, the absolute count of these switched DKO cells was almost negligible (*Figure 2—figure supplement 2D and E*). Such disruption in GC formation and IgG1 class-switching was also observed when DKO mice were challenged with another T-cell-dependent antigen, NP-CGG (*Figure 2G* and *Figure 2—figure supplement 2F and G*). Despite the compromised GC response, the frequencies and absolute numbers of total B220+CD19+B cells in the immunized DKO mice were comparable to their SKO and control littermates (*Figure 2F and H*). These experiments indicate that although *Gen1-Mus81*-null naïve B cells can persist in the periphery, they are unable to mount a productive GC reaction upon antigenic exposure at the barrier sites and in secondary lymphoid organs.

## GEN1 and MUS81 are necessary for B cell proliferation and survival

To mechanistically investigate the causes underlying the abrogated GC response in the DKO mice, we leveraged a tractable ex vivo culture system wherein purified splenic naïve B cells are induced to proliferate upon stimulation with various cocktails of mitogens and cytokines. The expansion of splenic B cell cultures stimulated with LPS alone, LPS+IL-4 (LI), or LPS+TGF-β+anti-IgD dextran (LTD) was monitored by enumerating the live cells in culture using flow cytometry. Across all stimulation conditions, the DKO B cells were unable to expand—after 96 hr of culture, the number of live DKO cells was only 10% of that of control and SKO B cells (*Figure 3A* and *Figure 3—figure supplement 1A*). We then examined the proliferation dynamics of DKO B cells by labeling the cells with CellTrace Violet and tracking the dilution of the dye over time (*Figure 3B*). We noted that at 72-hr post-stimulation, between 51% (LTD culture) and 68% (LPS culture) of the DKO cells had underwent at least three rounds of cell division, in contrast to between 72% (LTD culture) and 86% (LPS culture) of the corresponding control and SKO populations (*Figure 3C* and *Figure 3—figure supplement 1B*). Although the DKO LI and LPS cultures contained a comparatively higher frequency of undivided (division zero) cells, the absolute numbers of division zero cells in both stimulations were similar across all genotypes, supporting the notion that the accumulation of division zero cells reflects a disproportionate representation of undivided DKO cells in the cultures caused by the progressive attrition of proliferating DKO cells, rather than by the specific inability of DKO cells to initiate cell division (*Figure 3D* and *Figure 3—figure supplement 1C*). As cellular proliferation is intimately linked to class

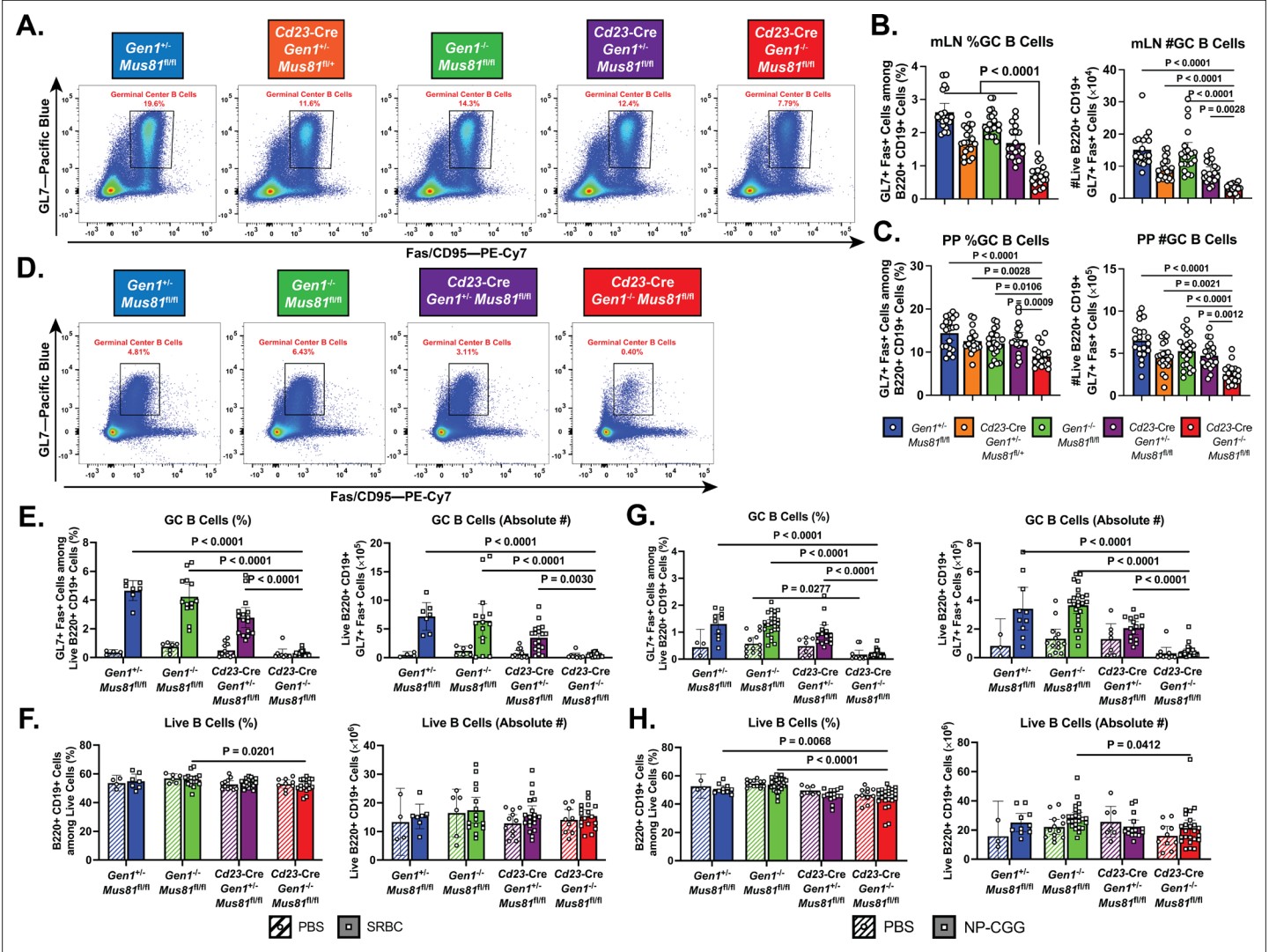

**Figure 2.** Homeostatic and induced GC responses in *Cd23*-Cre *Gen1*$^{-/-}$ *Mus81*$^{fl/fl}$ mice. (**A–C**) Homeostatic GC response in mesenteric lymph nodes and Peyer's patches. (**A**) Flow cytometric plots depicting the GL7+Fas+GC B cells in the Peyer's patches of mice for each indicated genotype. (**B, C**) Quantification of the frequencies and absolute numbers of the GL7+Fas+GC B cell population in the mesenteric lymph nodes (**B**) and Peyer's patches (**C**). (**D–F**) Evaluation of GC response during SRBC challenge. (**D**) Representative plots of the GL7+Fas+GC B cell population isolated from the spleen of mice immunized with SRBC for each indicated genotype. (**E, F**) Quantification of the frequencies and absolute numbers of GC B cells (**E**) and of total B cells (**F**) in the spleen of SRBC-immunized mice. (**G, H**) Assessment of induced GC response upon NP-CGG challenge in the *Cd23*-Cre *Gen1*$^{-/-}$ *Mus81*$^{fl/fl}$ mice at day 21 post-immunization. (**G**) Quantification of the percentage and absolute count of GL7+Fas+GC B cells in the spleen. (**H**) Frequencies and absolute numbers of live B220+ B cells in the spleen of PBS-treated and NP-CGG-immunized mice. Data in (**B**) and (**C**) are from four independent experiments with 11–21 mice per genotype. Data in (**E**) and (**F**) are from three independent experiments with 4–9 mice per genotype for the PBS group and 7–20 mice per genotype for the SRBC group. Data in (**G**) and (**H**) are from three independent experiments with 5–13 mice per genotype in the PBS group and 10–25 mice in the NP-CGG group. Bars represent the arithmetic mean and the error bars depict the 95% confidence interval of the measured parameters. For (**B**) and (**C**), p values were computed by ordinary one-way ANOVA analysis with Dunnett's multiple comparisons test without pairing in which the means were compared to the *Cd23*-Cre *Gen1*$^{-/-}$ *Mus81*$^{fl/fl}$ group. For (**E–H**), ordinary two-way ANOVA analysis with Dunnett's multiple comparisons test without pairing was used to calculate the p values. All means were compared within each treatment group to the *Cd23*-Cre *Gen1*$^{-/-}$ *Mus81*$^{fl/fl}$ cohort.

The online version of this article includes the following figure supplement(s) for figure 2:

**Figure supplement 1.** Steady-state phenotyping and quantification of the splenic and bone marrow (BM) B cell populations in the *Cd23*-Cre *Gen1*$^{-/-}$ *Mus81*$^{fl/fl}$ mice.

**Figure supplement 2.** Assessment of *Mus81* deletion efficiency in the germinal center by genomic qPCR and RT-qPCR analyses and examination of class switching in induced GCs in *Cd23*-Cre *Gen1*$^{-/-}$ *Mus81*$^{fl/fl}$ mice.

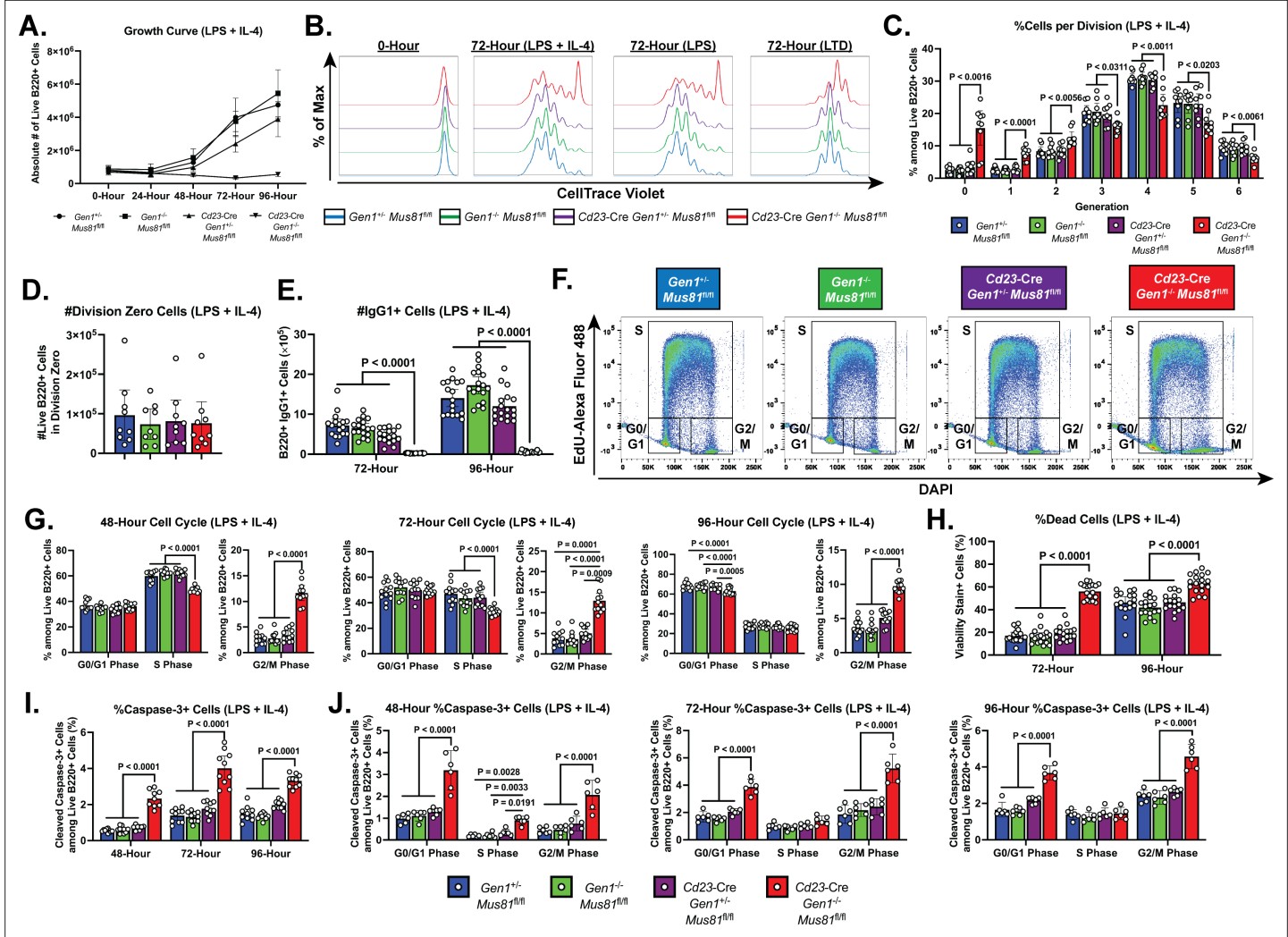

**Figure 3.** Growth, proliferation, cell cycle, and cell death profiles of ex vivo-stimulated DKO B lymphocytes. (**A**) Growth curve of LPS+IL-4 (LI)-activated B cell culture. (**B**) Representative CTV dilution profiles of ex vivo-activated B cells at 0- and 72-hr post-stimulation for all indicated genotypes and culture conditions. (**C**) Frequency of live B220+ cells in each division. (**D**) Absolute number of undivided (division zero) cells after 72 hr of LI culture. (**E**) Class switching of B lymphocytes to IgG1 at 72- and 96-hr post-activation quantified as absolute number of live IgG1-expressing B cells. (**F**) Representative flow cytometry plots delineating the cell cycle stages of the cultured B cells based on EdU positivity and nuclear DNA content as determined by the intensity of DAPI staining. (**G**) Frequency of live B cells in G0/G1, S, and G2/M phases after 48, 72, and 96 hours of activation. (**H**) Fraction of dead cells among B220+ singlets at 72 and 96 hr after LI activation. (**I**) Percentage of cleaved caspase-3+ cells among live cells at 48-, 72-, and 96-hr post-stimulation. (**J**) Frequency of cleaved caspase-3+ cells among live B220+ cells in G0/G1, S, and G2/M phases after 48, 72, and 96 hr of culture. Data in (**A**) are from four independent experiments with nine mice per genotype. Data in (**C**) and (**D**) are from four independent experiments with 7–9 mice per genotype. Data in (**E**) are from seven experiments with 17 mice per genotype. Data in (**G**) are from six independent experiments with 11–12 mice per genotype. Data in (**H**) are from seven experiments with 17 mice per genotype. Data in (**I**) are from five independent experiments with 9–10 mice per genotype. Data in (**J**) are from three independent experiments with six mice per genotype. Bars display the arithmetic mean and error bars represent the 95% confidence interval of the measured parameters. P values were determined using ordinary two-way ANOVA analysis with Dunnett's multiple comparisons test without pairing wherein the mean of the *Cd23*-Cre *Gen1*$^{-/-}$ *Mus81*$^{fl/fl}$ group was compared to the rest.

The online version of this article includes the following figure supplement(s) for figure 3:

**Figure supplement 1.** Ex vivo characterization of DKO cells following LPS and LPS+TGF-β+anti-IgD-dextran (LTD) stimulations.

**Figure supplement 2.** Cell death profiles of DKO cells stimulated in LPS and LTD cultures.

switching (*Hodgkin et al., 1996*; *Rush et al., 2005*), the lack of proliferative expansion of DKO B cells led to a minimal production of class-switched cells (*Figure 3E* and *Figure 3—figure supplement 1D*).

To gain additional insight into the proliferation defect sustained by the activated DKO cells, we analyzed the cell cycle profile of these cells. Cultured B cells were pulse labeled with the nucleoside

analog EdU to mark cells in S phase and stained with FxCycle Violet (DAPI) to quantify DNA content; the proportion of cells in G0/G1, S, and G2/M phases was determined by flow cytometry (*Figure 3F*). As early as 48-hr post-stimulation, the DKO cultures were enriched for cells in G2/M phase and depleted for cells in S phase (*Figure 3G*). This skewed cell cycle distribution of the DKO cells persisted up till 96-hr post-activation and was observed in all culture conditions (*Figure 3—figure supplement 1E and F*). Additionally, in the LTD culture, the fraction of DKO cells in the G0/G1 stage was reduced across all the time points examined (*Figure 3—figure supplement 1F*). Altogether, these results suggest that the concomitant loss of GEN1 and MUS81 causes the cells to stall in G2/M, impeding the completion of the cell cycle.

Aside from the proliferation deficiency, we also observed a 2- to 3-fold higher proportion of dead cells in the DKO cultures relative to that in control and SKO cultures (*Figure 3H* and *Figure 3—figure supplement 2A*). To ascertain whether elevated apoptosis contributed to the perturbed expansion of DKO B cell cultures, we quantified by flow cytometry the frequency of cells that stained for anti-cleaved caspase-3 antibody. As early as 48-hr post-stimulation, the proportion of caspase-3+ cells was 2- to 5-fold higher in DKO cultures compared with control and SKO cultures. The size of the caspase-3+ population peaked at 72-hr post-stimulation before declining (LI and LTD cultures) or remaining unchanged (LPS culture) at 96-hr post-stimulation (*Figure 3I* and *Figure 3—figure supplement 2B*). When we quantified the fraction of caspase 3+ cells in the different cell cycle phases, we found that across all the time points and stimulation conditions examined, DKO cells in G2/M and G0/G1 phases experienced a 2- to 5-fold higher level of apoptosis than their control and SKO counterparts (*Figure 3J* and *Figure 3—figure supplement 2C and D*). Taken together, these observations underscore an indispensable role for GEN1 and MUS81 in supporting the proliferative capacity and viability of activated B lymphocytes.

## Ablation of GEN1 and MUS81 induces p53 and type I interferon transcriptional programs

To assess the genome-wide transcriptional alterations underlying the proliferation and survival defects of ex vivo-activated DKO B cells, we conducted RNA-sequencing (RNA-seq) on activated B cells harvested at 48-hr post-stimulation, the time point at which the DKO cells were viable while displaying early signs of cell cycle perturbation and apoptosis. The RNA-seq analysis showed that the activated control and *Gen1*-KO B lymphocytes resembled each other transcriptomically, consistent with the lack of overt perturbations in *Gen1*-KO B cells (*Figure 4—figure supplement 1A*). Differential gene expression (DGE) analysis comparing *Mus81*-KO to control cells, however, identified 8 genes with a minimum of twofold upregulation in *Mus81*-KO cells (*Figure 4—figure supplement 1B*). The induction of only a few genes in *Mus81*-KO cells could be explained by the mild proliferation and survival perturbations that we had observed in our in vivo and ex vivo experiments. Between *Gen1*-KO and *Mus81*-KO cells, the transcript level of only four genes, including *Mus81,* were differentially altered (*Figure 4—figure supplement 1C*). These findings collectively illustrate that the deletion of either *Gen1* or *Mus81* alone does not substantially alter the transcriptional landscape of activated B lymphocytes.

DGE analysis of control versus DKO cells, on the contrary, revealed that 279 genes were upregulated by at least twofold ($\log_2$ fold change$\geq$1; FDR<0.05) and 167 genes were downregulated by a minimum of $\log_2$ fold change of $-0.3$ in the DKO B cells (corresponding to a $\geq$19% de-enrichment relative to control cells) (*Figure 4A*). To identify the functional modules to which the differentially expressed genes in the DKO cells belong, we performed gene set enrichment analysis (GSEA) with the Hallmark Gene Sets from the Molecular Signatures Database (MSigDB) (*Liberzon et al., 2015*). We identified the p53 pathway (25/200 genes with $\log_2$ fold change$\geq$1; e.g., *Ccng1, Zfp365, Plk2, Phlda3, Cdkn1a,* and *Zmat3*) and apoptosis (15/161 genes with $\log_2$ fold change$\geq$0.5) signatures among the top five enriched gene sets whereas gene sets containing targets of the transcription factors MYC and E2F and genes involved in progression through the G2/M checkpoint were among the most de-enriched in the DKO cells (*Figure 4B and C* and *Figure 4—figure supplement 1D-F*), concurring with our findings that DKO cultures exhibit perturbed cell cycle progression, G2/M arrest, proliferation irregularities, and heightened apoptosis. We also noted that the ex vivo-activated DKO B cells displayed a robust type I IFN gene signature, as exemplified by the high enrichment score (NES=2.35; FDR=6.34×10$^{-3}$) and the induction ($\log_2$ fold change$\geq$0.5) of 21/97 genes in the gene set (e.g., *Ifit3, Ifit3b, Ifitm3, Mx1,* and *Rsad2*) (*Figure 4C*). We posit from this analysis that the deficiency

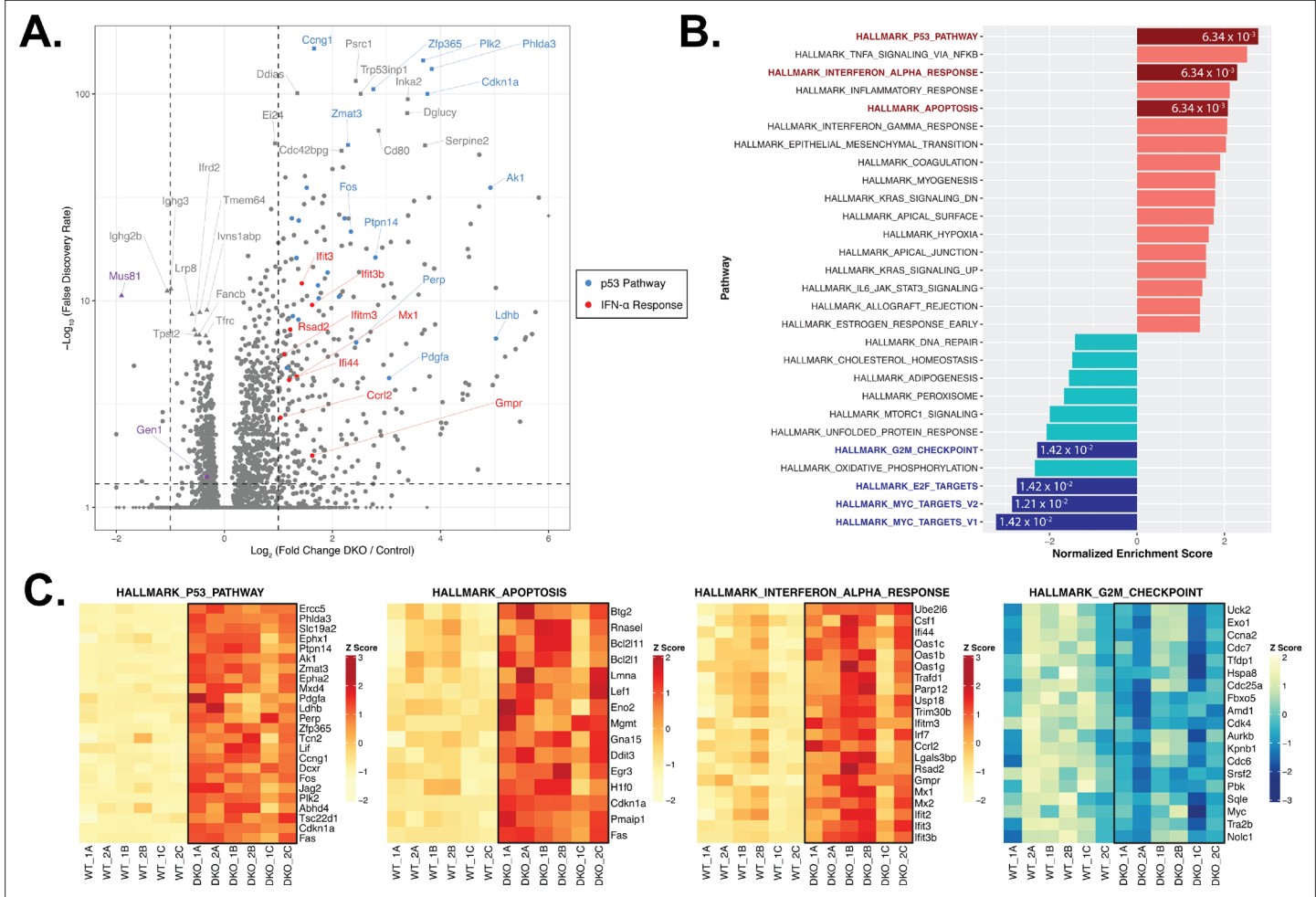

**Figure 4.** RNA-seq and gene set enrichment (GSEA) analyses of activated control and DKO B cells. (**A**) Volcano plot depicting the differential gene expression between control and DKO cells. Labeled squares indicate the top 15 most significantly upregulated genes and the labeled triangles are the 10 most downregulated genes in DKO cells. Using the Hallmark Gene Sets as reference, genes labeled in blue are categorized as genes in the p53 pathway while those labeled in red are defined as interferon alpha (IFN-α) response genes. Purple symbols mark *Gen1* and *Mus81*. (**B**) Graph depicting the list of Hallmark Gene Sets that are differentially expressed between control and DKO cells (FDR<0.05). Value in each bar denotes the FDR for that gene set. (**C**) Heatmaps displaying the relative expression of genes within the indicated Hallmark Gene Sets that meet the expression cutoff (log$_2$ fold change>1 for p53 pathway; >0.5 for apoptosis and interferon alpha response; <–0.3 for G2/M checkpoint; with FDR<0.05). Data are from six mice per genotype. The labels '1' and '2' represent male and female mice, respectively, and 'A' to 'C' indicate the three experimental ex vivo groups.

The online version of this article includes the following source data, source code, and figure supplement(s) for figure 4:

**Source code 1.** R script to generate *Figure 1A*, *Figure 4*, and *Figure 4—figure supplement 1*.

**Source code 2.** Bash script to clean and align FASTQ files.

**Source code 3.** Bash script to count gene alignments.

**Source data 1.** Differential expression between control and DKO B cells.

**Figure supplement 1.** Transcriptomics and gene set enrichment (GSEA) analyses of ex vivo-activated control, GKO, and MKO B cells.

**Figure supplement 1—source data 1.** Differential expression between control and GKO B cells.

**Figure supplement 1—source data 2.** Differential expression between control and MKO B cells.

**Figure supplement 1—source data 3.** Differential expression between GKO and MKO B cells.

**Figure supplement 2.** In vivo and ex vivo phenotyping of DKO *Trp53*-WT, DKO *Trp53*-Het, and DKO *Trp53*-KO B cells.

of both GEN1 and MUS81 in cells activates p53-dependent pathways to arrest cell cycle for the repair of genomic insults sustained during cellular growth and proliferation, and to initiate apoptosis when such DNA lesions are beyond tolerance and repair.

## p53 deficiency fails to ameliorate the proliferative and viability defects of GEN1-MUS81-double-null B cells

Since the DKO cells exhibit a strong p53 transcriptional signature that could drive their apoptosis and G2/M arrest, we wanted to determine whether the deletion of p53 would mitigate these detrimental events and restore the growth and functionality of DKO cells. To test this hypothesis, we bred *Trp53*$^{fl/fl}$ mice to the DKO mouse to generate *Cd23*-Cre: *Gen1*$^{-/-}$ *Mus81*$^{fl/fl}$ *Trp53*$^{fl/fl}$ (DKO *Trp53*-KO) mice.

Examination of the GCs in the mLN and Peyer's patches of DKO *Trp53*-KO mice and their p53-expressing littermates (DKO *Trp53*-WT and DKO *Trp53*-Het) revealed no difference in the absolute number of total B cells and in the frequency of GC B cells among these groups (*Figure 4—figure supplement 2A and B*). Likewise, growth curve and CellTrace Violet dilution analyses showed no alterations in the proliferative competence of DKO *Trp53*-KO B cells (*Figure 4—figure supplement 2C and D*). These cells, however, did exhibit small but incongruent differences in their class-switching capacities (*Figure 4—figure supplement 2E*). Further, the cell cycle profile of the DKO *Trp53*-KO cells was not substantively different from that of the DKO *Trp53*-WT and DKO *Trp53*-Het cells (*Figure 4—figure supplement 2F*).

Despite the comparable percentage of dead DKO cells independent of their p53 status (*Figure 4—figure supplement 2G*), the DKO *Trp53*-KO B cells underwent reduced level of apoptosis at 48-hr post-stimulation compared with their p53-expressing counterparts (*Figure 4—figure supplement 2H*). This decrease was not sustained at subsequent time points; instead, the fraction of apoptotic DKO *Trp53*-KO cells increased and became higher than that of p53-expressing DKO B cells (*Figure 4—figure supplement 2H*). Assessment of apoptosis in the individual cell cycle phase showed that the temporal changes in the cell death of p53-null DKO cells occurred consistently in the G2/M phase (*Figure 4—figure supplement 2I*). These data together show that the severe proliferative and survival deficiencies of DKO B cells cannot be overcome by the removal of p53.

## GEN1 and MUS81 maintain the genome stability of activated B lymphocytes

GEN1 and MUS81 resolve structural intermediates arising from replication and HR to prevent genome destabilization caused by the toxic accumulation of erroneously processed branched and joint DNA structures (*Chan et al., 2018*; *Sarbajna et al., 2014*). Reasoning that the genomic integrity of proliferating DKO B cells might similarly be compromised, we prepared metaphase spreads from ex vivo-stimulated B cells for evaluation of chromosomal integrity. DKO B cells displayed approximately six times as many abnormalities per metaphase as control and SKOs B cells (*Figure 5A and B*). Furthermore, whereas almost 95% of control and SKO metaphases had no more than two aberrations, 45% of DKO metaphases showed three or more chromosomal defects (*Figure 5C*). Chromosomal aberrations, including breaks, DNA fragments, fusions, and radials, were detected at an elevated rate in DKO B cells compared with control and SKOs cells (*Figure 5D*). Notably, 67% of the breaks observed were chromosome-type breaks that occur at the same position on both sister chromatids (*Figure 5E*). To better understand the origins and nature of the chromosomal irregularities occurring in the activated DKO B cells, we performed telomere fluorescence in situ hybridization (Tel-FISH). In DKO cells, the breakage occurred proximal to the telomeres, resulting in paired DNA fragments containing telomeric DNA (*Figure 5F*). Such symmetrical breakage has been proposed to arise from unresolved recombination intermediates (*Garner et al., 2013*). Additionally, the Tel-FISH analysis revealed that the aberrations identified by Giemsa staining as acentric chromosomes in fact often involved two separate chromosomes, and thus likely represent a class of radials involving acrocentric short arms (*Figure 5—figure supplement 1*). These observations show that the absence of GEN1 and MUS81 engenders an assortment of chromosomal lesions in proliferating B cells.

Because AID-instigated DSBs within and outside of the *IgH* loci constitute a source of DNA damage in B lymphocytes, we generated *Cd23*-Cre: *Gen1*$^{-/-}$ *Mus81*$^{fl/fl}$ *Aicda*$^{-/-}$ (DKO *Aicda*-KO) mice to ascertain whether ablation of AID can rescue the viability and proliferation defects of DKO B cells. Ex vivo culture studies showed that the deletion of AID did not restore the proliferative and survival capacities

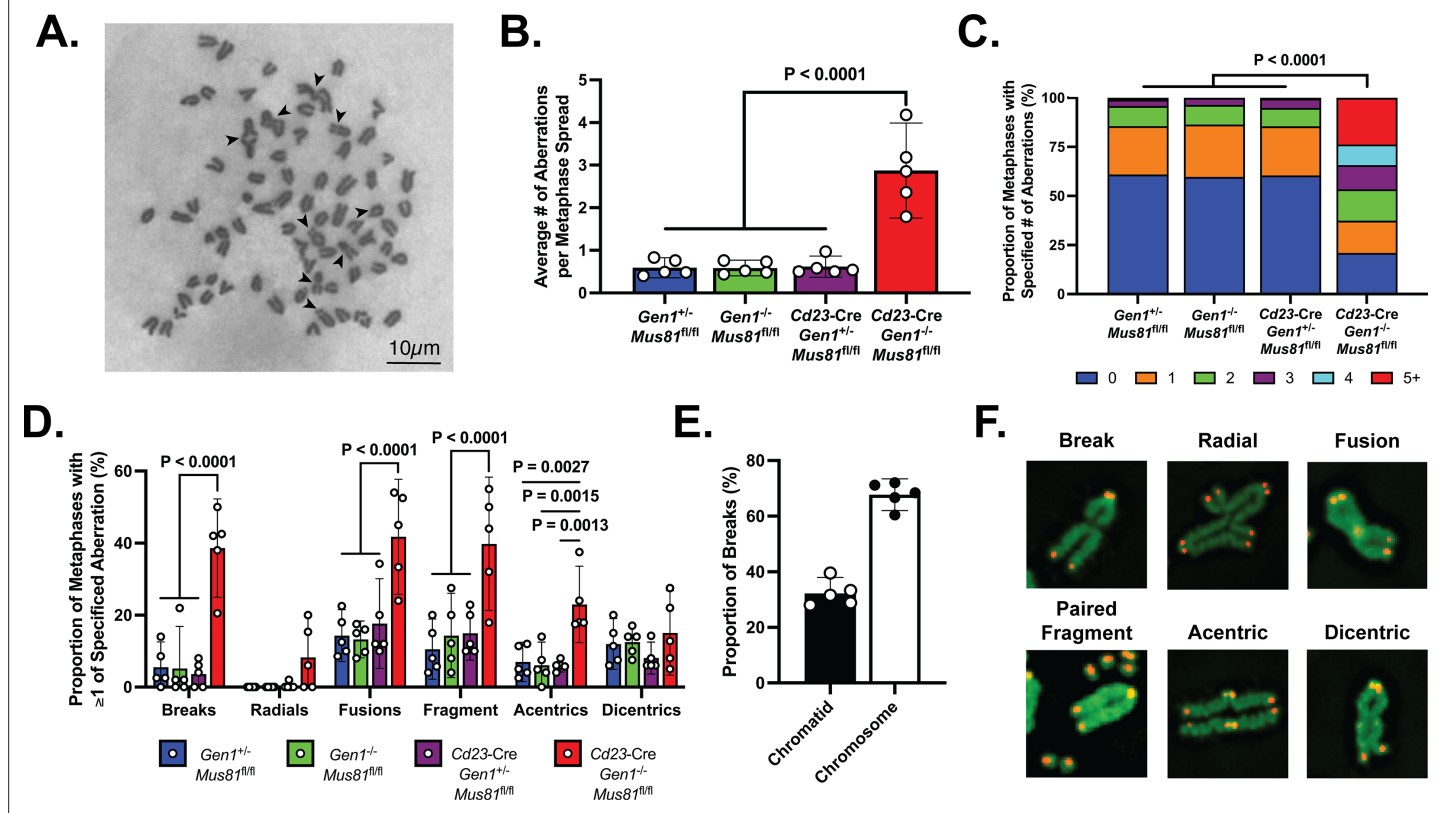

**Figure 5.** Metaphase chromosomal analysis of activated DKO B cells. (**A**) Representative image of a DKO metaphase spread with arrows indicating chromosomal breaks, fragments, and fusions in metaphases of activated DKO B cells. (**B**) Quantification of the average number of chromosomal aberrations across 45–50 metaphase spreads prepared from each B cell culture. (**C**) Percentage breakdown of metaphases exhibiting 0 to greater than 5 chromosomal aberrations. (**D**) Fraction of metaphases containing the indicated types of chromosomal abnormalities. Total percentage per genotype exceeds 100% as some metaphases exhibit more than one type of abnormality. (**E**) Proportion of chromatid and chromosome breaks among the 163 breaks observed in DKO metaphase spreads exhibiting at least one break. (**F**) Tel-FISH images of DKO metaphases highlighting the proximal location of the chromosomal damage to the telomeres. Note that the events labeled as dicentrics here and in *Figure 5—figure supplement 1* may include chromosomes with residual cohesins remaining at a repaired DSB and those with condensins that persist after loading onto the chromosomal arms during mitotic entry. Data in (**B–E**) are from three independent experiments with 5 mice (totaling between 215 and 235 metaphase spreads) per genotype. For (**C**), the percentages are the average of the data combined from all five mice. Bars display the arithmetic mean and error bars represent the 95% confidence interval of the measured parameters. P values were computed with ordinary one-way ANOVA analysis (**B, D**) and the Kruskal-Wallis test (**C**) with Dunnett's multiple comparisons test without pairing. Means of all groups were compared to that of *Cd23*-Cre *Gen1*$^{-/-}$ *Mus81*$^{fl/fl}$.

The online version of this article includes the following figure supplement(s) for figure 5:

**Figure supplement 1.** Telomere FISH analysis of metaphase spreads prepared from ex vivo-activated DKO B cells.

**Figure supplement 2.** Proliferation dynamics, viability, and chromosomal analyses of DKO *Aicda*-Het and DKO *Aicda*-KO ex vivo cultures.

of DKO cells (*Figure 5—figure supplement 2A-D*). Additionally, the average number of chromosomal aberrations in the DKO *Aicda*-KO cells was similar to that of DKO *Aicda*-Het B cells (*Figure 5—figure supplement 2E*). These findings illustrate that AID-driven DSBs contribute little to the severe proliferation and viability defects exhibited by *Gen1-Mus81*-null B lymphocytes and that the chromosomal aberrations occurring within the DKO B cells are likely caused by unrepaired replication-associated DNA damage.

## Discussion

B cells encounter a diverse array of genotoxic stresses throughout their life cycle, of which DSBs—both spontaneously and deliberately generated—are considered among the most deleterious genomic lesions. HR constitutes one of the two major pathways cells utilize to repair DSBs. This mode of repair is largely restricted to late S and G2 phases as it requires the sister chromatid as a template to

restore the fidelity of the damaged DNA strand (*Heyer, 2015*). Recombinational repair of DSB lesions produces various joint and branched molecules that must be processed accurately and timely by SSEs to maintain the integrity of the genome (*Blanco and Matos, 2015*; *Dehé and Gaillard, 2017*). Due to their broad substrate specificity, SSEs are stringently regulated within a physiological environment via multiple mechanisms including binding with regulatory proteins and post-translational modifications that constrain their in vivo target recognition, thus endowing them with context-specific cellular functions (*Wild and Matos, 2016*). To examine the genome-stabilizing role of GEN1 and MUS81 at various stages of B cell development, we established a *Gen1*$^{-/-}$ *Mus81*$^{fl/fl}$ mouse model and utilized early and late B-cell-specific Cre drivers for the conditional deletion of MUS81. Inactivation of both enzymes in early B cell precursors abolished the production of mature B cells in the BM and periphery, whereas ablation in mature, naïve B cells impaired GC formation in response to antigenic stimulation. Ex vivo cellular and transcriptomics analyses revealed that these endonuclease-deficient B cells exhibited significant proliferation and viability perturbations underpinned by widespread chromosomal abnormalities and activation of cell cycle arrest and apoptosis in response to protracted p53 signaling elicited by extensive DNA damage.

The severe attrition of pro-B cells in the *Mb1*-Cre *Gen1*$^{-/-}$ *Mus81*$^{fl/fl}$ mice could be caused by a developmental arrest during the transition from pre-pro B cell to pro-B cell stage, by proliferation and survival defects of the pro-B cells, or both. Pro-B cells, compared to the mitotically inactive pre-pro B cells, undergo IL-7R-dependent proliferative expansion prior to V(D)J recombination and so are more likely to suffer from intense replication stress that could necessitate commensurate effort of replication restart (*Hardy et al., 1991*; *Peschon et al., 1994*). As the SSE-knockout pro-B cells are unable to resolve both stalled replication forks and recombination-associated DNA structures formed during such restart activities, they would fail to thrive and proliferate further. Similarly, the lack of robust chronic and induced GC responses in the *Cd23*-Cre *Gen1*$^{-/-}$ *Mus81*$^{fl/fl}$ mice can be ascribed to the inability of DKO cells to alleviate replication stress generated during sustained proliferation by eliminating replication-induced junction molecules, leading to G2/M stalling and abrogation of a complete cell division. High levels of concurrent DNA replication and transcriptional activities within the GC B cells could promote replication-transcription conflicts and R-loop formation that further exacerbate replication stress. We did not observe a time-dependent G2/M accumulation in the DKO culture, suggesting that the arrested DKO cells could not tolerate for long periods the high level of DNA damage incurred from multiple rounds of cell division and consequently undergo mitotic catastrophe, evidenced in the comparatively higher rate of apoptosis among cells in the G2/M phase. Resting mature DKO cells, however, persisted unscathed in the spleen, as exemplified by the normal frequencies and numbers of splenic B cell subpopulations in the *Cd23*-Cre *Gen1*$^{-/-}$ *Mus81*$^{fl/fl}$ mice. Splenic naïve B cells do not self-renew; instead, they are constantly replenished from immature precursors produced in the BM and maintained homeostatically by the survival signals BAFF and APRIL (*Hao and Rajewsky, 2001*; *Mackay and Gommerman, 2015*). Hence, resting mature B cells are spared from replication-derived genotoxicity, rendering GEN1 and MUS81 dispensable in these cells.

The inability of the BTR complex to compensate for the concomitant deficiencies of GEN1 and MUS81 and maintain the viability of proliferating DKO B cells suggests that DNA replication and recombination events generate persistent double HJs and multiple BTR-refractory HJ species such as single HJs and nicked HJs that must be resolved by these endonucleases to suppress genomic instability and catastrophic mitosis (*García-Luis and Machín, 2014*). Nevertheless, the absence of these SSEs may also preclude the processing of aberrant replication intermediates that could deleteriously impact the genome duplication process and undermine the proliferative capacity of the cells. In particular, the MUS81-EME2 complex has been shown to mediate replication fork restart by endonucleolytic cleavage of stalled replication forks during S phase to protect genome integrity (*Pepe and West, 2014a*). Although the in vivo relevance of GEN1 in processing such intermediates remains to be formally established, our genetic analysis and others suggest that when MUS81 is unavailable, GEN1 can potentially act on these substrates (or a processed form of them) that persist into mitosis upon nuclear breakdown to initiate HR-dependent fork restart (*García-Luis and Machín, 2014*; *Ho et al., 2010*; *Sarbajna et al., 2014*). Deregulated mutagenic synthesis of the DNA strand during BIR can also contribute to the genomic instability of the DKO cells. Deficiency in these SSEs enables the uncleaved D-loop to undergo prolonged extension, promoting mutagenesis, template switching, and nonreciprocal loss of heterozygosity (*Ho et al., 2010*; *Mayle et al., 2015*). SSEs can presumably act on nascent

D-loops to promote the less deleterious genetic outcome of half crossovers and on extended D-loops to restrict error-prone DNA synthesis (*Mayle et al., 2015*; *Schwartz and Heyer, 2011*). Alternatively, the inability to eliminate single HJ intermediates formed from the merger of an incoming replication fork with the extending D-loop may explain the genotoxicity of BIR in *Gen1-Mus81*-DKO cells (*Mayle et al., 2015*). Although the disruption of the D-loop is central to the initiation of SDSA, no experimental evidence has emerged thus far to support such a role by MUS81, and we believe this to be highly unlikely due to its nicking mechanism. The MUS81 complexes, nonetheless, may be implicated in facilitating SDSA because of their 3′-flap cleavage activity (*Hollingsworth and Brill, 2004*).

Our observation of extensive chromosomal aberrations in the endonuclease-null mouse B cells concur with that of *Sarbajna et al., 2014* that employed siRNA depletion of *Gen1* and *Mus81* in cells treated with replication inhibitors. Recombination intermediates generated in S phase that fail to be resolved in the absence of GEN1 and MUS81 evade the checkpoint response and persist into mitosis, generating homologous recombination ultrafine bridges (HR-UFBs) (*Chan et al., 2018*; *Mohebi et al., 2015*; *Tiwari et al., 2018*). Breakage of the HR-UFBs during cytokinesis engenders chromosomal breaks that activate the DNA damage checkpoint in the next cell cycle, triggering non-homologous end joining-mediated fusion of DNA ends that leads to widespread chromosomal rearrangements (*Chan et al., 2018*). The preponderance of chromosome breaks that appear to occur at identical sites on sister chromatids suggest that the aberrations result from defective resolution of inter-chromatid recombination intermediates (*Kikuchi et al., 2013*; *Shimizu et al., 2020*; *Wechsler et al., 2011*). This phenotype is distinct from that of irradiated HR-defective mutants in which chromatid breaks predominate (*Fujita et al., 2013*; *Shimizu et al., 2020*). Unresolved HJs have been implicated to impede DNA condensation in mitosis, manifesting as 'pinches' in the chromosome (*Wechsler et al., 2011*); thus, a subset of the chromosome breaks observed in DKO cells may not be actual breaks but rather intact HJs. Although these chromosome-type breaks may also arise due to defective NHEJ, the established roles of GEN1 and MUS81 in HJ resolution render this explanation unlikely. We also cannot completely exclude the role of MUS81—and potentially GEN1—in cleaving stalled replication forks to initiate BIR and MiDAS at CFS and other under-replicated regions where such failure leads to the formation of FANCD2-flanking fragile site-UFBs (FS-UFBs) between the segregating chromatids (*Naim et al., 2013*; *Ying et al., 2013*). However, the frequent occurrence of breaks at corresponding locations on paired sister chromatids argues for a failure of DKO B cells to resolve HR intermediates. Visualization of UFBs in these cells hence may help delineate the molecular origins of these unique aberrations.

Detailed investigations into whether HR-UFBs derive predominantly from genomic loci where stalled replication forks are preferentially restarted through recombination-dependent mechanisms could provide mechanistic insights into the genomic instability of SSE-null B cells and clarify the manner by which GEN1 and MUS81 resolve replication stress—do they cleave persistent and late-occurring replication intermediates such as reversed forks or do they process HR intermediates (e.g., D-loops and HJs) arising from recombination-mediated restart of perturbed forks? Further studies to determine whether the loci where HR-UFBs manifest encompass early replication fragile sites (ERFSs) should be pursued given that ERFSs exhibit higher level of HR-dependent sister chromatid exchanges than CFSs upon chemical induction of replication stress (*Waisertreiger et al., 2020*). More importantly, such sites have recently been identified as hotspots for transcription-replication conflicts and breakpoints of chromosomal rearrangements in B lymphocytes; hence, elucidating the molecular events involved in rectifying replication stress at ERFSs will be critical to better understanding the drivers of genomic instability (*Barlow et al., 2013*).

The robust transcriptional upregulation of p53 signaling and apoptosis gene programs coupled with the de-enrichment of gene sets related to G2/M checkpoint progression, E2F targets, and MYC targets in activated DKO cells is consistent with the cell cycle and survival perturbations exhibited by these cells. Cellular stresses including DSBs trigger the p53 stress response that entails global transcriptional inhibition with activation of specific gene programs to induce metabolic rewiring, DNA damage repair, G1 and G2 arrests, apoptosis, and senescence (*Levine, 2020*). Although the loss of GEN1 and MUS81 has not been directly linked to the downregulation of MYC signaling, the DNA damage response activated in the DKO cells causes the accumulation of p53 protein, resulting in elevated p53 binding to an enhancer element within a MYC super-enhancer and consequent repression of *Myc* mRNA transcription (*Porter et al., 2017*). Moreover, p53 can directly impact the expression of *Myc* via other transcriptional and post-transcriptional mechanisms. p53 can activate the transcription of the

long noncoding RNA *Pvt1b* to suppress in cis *Myc* transcription (*Olivero et al., 2020*). miR-34 family and miR-145 are transcriptional targets of p53 and they contribute to the silencing of *Myc* expression by eliciting the degradation of Myc mRNA transcripts (*He et al., 2007*; *Sachdeva et al., 2009*). Other transcriptional targets of p53, including PTEN, TSC2, and AMPKβ, can impinge on MYC expression via their inhibition of the upstream regulator mTORC1 complex (*Cui et al., 2021*; *Levine, 2020*). These MYC-counteracting mechanisms are crucial to the proper enforcement of p53-induced cell cycle arrest and apoptosis. Our genetic study showed that the loss of p53 was insufficient to rescue the abrogated GC response and proliferative expansion of DKO B cells. Though p53 deletion did prevent apoptosis of ex vivo-activated DKO B cells early on, this survival benefit was only temporary as the p53-null DKO B cells experienced much higher level of cell death at later time points. We infer that the absence of p53 precludes the DKO p53-KO cells from arresting their cell cycle despite incurring extensive replication-associated DNA damage, leading to mitotic catastrophe and their eventual demise.

Our study presents the first evidence that the ablation of both GEN1 and MUS81 induces the expression of interferon-stimulated genes (ISGs), most likely attributed to the elevated levels of micronuclei in the DKO cells (*Sarbajna et al., 2014*). Primary human fibroblast cells lacking the BLM dissolvase exhibit an enhanced ISG gene expression signature due to the increased level of cytoplasmic DNA and cGAS-positive micronuclei in these cells, suggesting a common transcriptional outcome in cells that fail to eliminate recombination-associated structures (*Gratia et al., 2019*). Micronuclei are formed when chromosomal fragments or lagging chromosomes—manifestations of mitotic or DNA repair defects—become enveloped in a rupture-prone nuclear membrane after failing to be incorporated into the nucleus after mitosis (*Miller et al., 2021*). Recognition of the micronuclei by cyclic GMP-AMP synthase (cGAS) triggers the catalytic production of the second messenger cyclic 2′3′-GMP-AMP (2′3′-cGAMP), promoting the phosphorylation of the adaptor stimulator of interferon genes (STING) that mediates the activation of interferon-regulatory factor 3 (IRF3) to drive the transcription of ISGs (*Harding et al., 2017*; *Li and Chen, 2018*; *Mackenzie et al., 2017*). Production and release of chromatin fragments from spontaneously ruptured micronuclei and from improper degradation of nascent DNA at unprotected, stalled replication forks further contribute to the cytosolic pool of immunostimulatory DNA fragments (*Coquel et al., 2018*; *Dou et al., 2017*; *Emam et al., 2022*; *Glück et al., 2017*; *Ragu et al., 2020*). DNA-stimulated type I IFN signaling, besides promoting the synthesis and secretion of proinflammatory cytokines, triggers various cell death programs including apoptosis, necroptosis, and autophagic cell death (*Paludan et al., 2019*). Recent evidence in T cells, however, suggests that p53 signaling can be directly activated by STING via cyclic dinucleotides to trigger cell cycle arrest and apoptosis independent of type I IFN production (*Concepcion et al., 2022*). We speculate that the cytoplasm of the DKO B cells could accumulate a panoply of self-DNA materials that fosters the production of 2′3-cGAMP and type I IFNs, initiating multiple cell death pathways simultaneously. This raises the question as to whether the cGAS-STING cascade comprises a secondary cell death signaling axis in the SSE-null cells, and whether micronucleated DKO B cells could elicit cell death in neighboring cells through the paracrine release of 2′3′-cGAMP and type I IFNs, given that macrophages and T cells can import 2′3′-cGAMP via the heteromeric anion channels LRRC8A-LRRC8E and LRRC8A-LRRC8C, respectively, to activate STING and type I IFN production (*Concepcion et al., 2022*; *Zhou et al., 2020*). Future studies examining the impact of cGAS or STING inhibition on the viability DKO B cells will be warranted to assess the role of these factors in DNA-stimulated intrinsic and extrinsic cell death pathways.

We posit that the illegitimate processing of branched DNA structures and the persistence of unresolved HR intermediates generated following replication fork stalling and recombination-mediated fork restart in proliferating B cells lacking GEN1 and MUS81 lead to the manifestation of aberrant mitotic structures including chromatin bridges, HR-UFBs, and micronuclei that enable rampant fusions of broken chromosomes and amplification of replication-derived DNA damage during the next cell cycle. Our studies show that such extensive chromosomal instability and genomic damage consequently activate p53-dependent G2/M arrest and apoptosis. Concurrently, these endonuclease-deficient B cells exhibit a type I IFN transcriptional signature, potentially a ramification of high levels of cytoplasmic self-DNA. The synthetic lethality of GEN1 and MUS81 deficiencies in B cells highlights the essentiality of SSEs in eliminating branched and joint DNA intermediates formed during replication, recombination, and repair events to safeguard the genomic integrity of proliferating cells and to

confer the proliferative and survival capacities required for the proper development and functionality of B lymphocytes, and possibly other immune cells.

## Materials and methods
### Mice

*Gen1*−/− mice were generated at the Memorial Sloan Kettering Cancer Center Mouse Genetics Core Facility. *Mus81*fl/fl mice were generated using ES cell clone purchased from EUCOMM (*Skarnes et al., 2011*). *Mb1* (*Cd79a*)-Cre mice were purchased from The Jackson Laboratory (strain #:020505) (*Hobeika et al., 2006*). *Cd23*-Cre (*Kwon et al., 2008*) and *Aicda*−/− (*Muramatsu et al., 2000*) mice were gifted by Meinrad Busslinger (Research Institute of Molecular Pathology, Austria) and Tasuku Honjo (University of Kyoto, Japan), respectively. *Trp53*fl/fl mice were purchased from The Jackson Laboratory (strain #:008462) (*Marino et al., 2000*). Experiments were performed using mice between 8- and 16-week-old. When littermate controls were unavailable, age-matched controls were employed in experiments. All mice were housed and maintained in groups of five under specific pathogen-free conditions, and euthanized at the time of analyses in accordance with guidelines for animal care established by Memorial Sloan Kettering Cancer Center Research Animal Resource Center and the Institutional Animal Care and Use Committee (IACUC).

### Flow cytometry and cell sorting

Single-cell suspensions were prepared from mouse spleen, mesenteric lymph nodes, and Peyer's patches by pressing through a 70 µm cell strainer (Corning), and BM cells were harvested by flushing the tibia with B cell media (RPMI 1640 with L-glutamine (Gibco) supplemented with 15% fetal bovine serum (Corning), 1% penicillin-streptomycin (GeminiBio), 2 mM L-glutamine (Memorial Sloan Kettering Cancer Center Media Preparation Facility), and 55 µM β-Mercaptoethanol (Gibco)). Splenic and BM suspensions were resuspended in red blood cell lysis buffer (150 mM $NH_4Cl$, 10 mM $KHCO_3$, and 0.1 mM EDTA) for 5 min at room temperature and then neutralized with B cell media. After washing with $Ca^{2+}$- and $Mg^{2+}$-free phosphate-buffered saline (PBS), cells were stained with Zombie Red fixable viability dye (BioLegend) and rat anti-mouse CD16/CD32 Fc Block (BD Biosciences) at room temperature for 15 min in the dark, followed by staining with antibodies for cell surface markers at 4°C for 30 min in the dark. The following antibodies and their respective clones were used in this study: B220 (RA3-6B2), CD19 (ID3), TCRβ (H57-597), CD43 (R2/60), IgD (11.26.2a), IgM (II/41; polyclonal), CD25 (PC61), CD249 (BP-1), c-Kit (2B8), CD93 (AA4.1), CD24 (M1/69), CD138 (281-2), CD21/CD35 (7E9), CD23 (B3B4), GL7 (GL7), CD95 (Jo2), CD38 (90), IgG1 (X56), IgG3 (R40-82), and IgA (mA-6E1). For the sorting of pro-B (fractions B+C) cells, the antibodies used for the dump channel were TCRb (H57-597), Cd11b (M1/70), Ter-119 (TER-119), NK1.1 (PK136), and Gr1 (RB6-8C5). All antibodies were purchased from BD Biosciences, eBioscience, and BioLegend. For intracellular cleaved caspase-3 staining, ex vivo-stimulated cells were incubated in Zombie Red fixable viability dye and rat anti-mouse CD16/CD32 Fc Block, stained with antibodies for cell surface antigens, followed by staining with anti-cleaved caspase-3 antibody (C92-605; BD Biosciences) for 45 min at 4°C after processing with Fixation/Permeabilization kit (BD Biosciences) according to the manufacturer's protocol. Data were obtained using an LSR II flow cytometer (BD Biosciences) and analyzed with FlowJo 10.6 (BD Biosciences). Pro-B (fractions B+C), GC, and non-GC B cells were sorted using a BD FACS Aria II (BD Biosciences).

### Immunization

For SRBC immunization, packed SRBCs (Innovative Research) were washed three times with PBS, counted with hemocytometer, and resuspended to a concentration of 10 million cells/µl. About 500 million cells were then administered intraperitoneally. Mice were boosted with the same number of SRBCs on day 10 before spleens were harvested for analysis on day 14. For NP-CGG immunization, mice were injected intraperitoneally with 100 µg NP-CGG (ratio 30–39; Biosearch Technologies) resuspended in Imject Alum adjuvant (Thermo Fisher Scientific). Mice were boosted on day 14 and euthanized on day 21 for analysis of immune response in the spleen.

## Primary B cell ex vivo stimulation

Splenic B cells were harvested and processed into single-cell suspensions by pressing through a 70 µm cell strainer. Naïve B cells were then purified by negative selection using anti-CD43 microbeads (Miltenyi Biotec) according to the manufacturer's protocol. B cells were plated at a density of 1×10⁶ cells/ml in B cell media in a six-well dish. B cells were then stimulated with one of the following cytokine cocktails: 33 µg/ml LPS (Sigma-Aldrich); 33 µg/ml LPS plus 25 ng/ml IL-4 (R&D Systems); or 10 µg/ml LPS, 2 ng/ml recombinant human TGF-β1 (R&D Systems), and 333 ng/ml anti-IgD dextran conjugates (Fina Biosolutions). Cultures were split by half at 48- and 72-hr post-stimulation.

## RT-qPCR

Total RNA was harvested from ex vivo B cell cultures, sorted pro-B (fractions B+C), and sorted GC and non-GC B cells using Quick-RNA Microprep Kit (ZymoResearch), Arcturus PicoPure RNA Extraction Kit (Thermo Fisher Scientific), and AllPrep DNA/RNA Micro Kit (QIAGEN), respectively, and subsequently reverse transcribed to cDNA using High-Capacity cDNA Reverse Transcription Kit (Applied Biosystems). TaqMan probes specific for *Gen1* (Mm00724023_m1), *Mus81* (Mm00472065_m1), and *Ubc* (Mm01201237_m1) were used to amplify the cDNA transcripts. qPCR experiments were performed with TaqMan Fast Advanced Master Mix (Applied Biosystems) in a 384-well format using an Applied Biosystems QuantStudio 6 Flex instrument. Relative gene expression was calculated using the $2^{-\Delta\Delta CT}$ method and normalized to *Ubc* expression.

## Genomic qPCR

Genomic DNA was isolated from sorted pro-B (fractions B+C) and from sorted GC and non-GC B cells using Arcturus PicoPure DNA Extraction Kit (Thermo Fisher Scientific) and AllPrep DNA/RNA Micro Kit (QIAGEN), respectively, according to the manufacturer's protocol. Primer pairs targeting the $Mus81^{fl/fl}$ allele (Fwd: 5′- CCGGAACCGAAGTTCCTATT-3′ and Rev: 5′- GTACAAGAAAGCTGGGTCTAGATA-3′) and *Ubc* (Fwd: 5′-AGTCGCCCGAGGTCACA-3′ and Rev: 5′-CGTCTCTCTCACGGAGTT GTTT-3′) were used to amplify the genomic DNA. Experiments were performed with PowerUp SYBR Green Master Mix (Applied Biosystems) in a 384-well format using an Applied Biosystems QuantStudio 6 Flex instrument. Relative copy number of the unrecombined $Mus81^{fl/fl}$ allele was determined using the $2^{-\Delta\Delta CT}$ method and normalized to *Ubc* copy number.

## Proliferation analysis

Purified naïve splenic B cells were stained with 5 µM CellTrace Violet (Invitrogen) in PBS for 20 min at room temperature in the dark. Cells were washed with B cell media to quench the dye before resuspension in fresh B cell media and subsequent incubation for at least 10 min at 37°C. Equal labeling between the genotypes was verified by flow cytometry immediately after labeling. Cytokine cocktails were then added to the B cell cultures to initiate stimulation.

## Cell cycle analysis

Prior to flow cytometric analysis, ex vivo B cells were treated with 10 µM EdU for 1 hr. Cells were harvested and washed with PBS before staining with antibodies for surface proteins. Cells were then processed using Click-iT Plus EdU Alexa Fluor 488 Flow Cytometry Assay Kit (Invitrogen) according to the manufacturer's protocol. Cells were subsequently stained with FxCycle Violet Stain (Invitrogen) for 30 min at room temperature in the dark before flow cytometry.

## RNA-sequencing library generation and analyses

B cells were cultured for 48 hr before total RNA was extracted using Quick-RNA Microprep Kit (ZymoResearch) and mRNA was isolated using the NEBNext Poly(A) mRNA Magnetic Isolation (New England BioLabs). Stranded Illumina libraries were prepared with Swift Rapid RNA Library Kit according to the manufacturer's instructions (Swift Biosciences). Indexed libraries were sequenced on a HiSeq X Ten platform, and an average of 30 million 150 bp paired-end reads were generated for each sample (Novogene, Beijing, China). The resulting FastQ files were processed to remove adapters and low-quality reads using GATK v4.1.9.0 (*Van der Auwera et al., 2013*). STAR v2.7.7a (*Dobin et al., 2013*) aligned the reads to GRCm38.p6 and gencode vM25 (*Frankish et al., 2021*), and GATK removed the duplicates. A count matrix was generated using featureCounts v2.0.1 (*Liao et al., 2014*), and DESeq2 v1.30.1 (*Love et al., 2014*) generated differential expression matrices. ggPlot2 v3.3.4

(*Wickham, 2016*) was used for creating volcano plots, highlighting genes that fall in the designated areas (see text). fgsea v1.16.0 (*Korotkevich et al., 2021*) and msigdb h.all.v7.4 (*Liberzon et al., 2015*; *Subramanian et al., 2005*) analyzed the differentially expressed genes (those with FDR<0.05) in DKO cells compared to control cells to determine which gene sets were enriched and de-enriched. Then, ggplot2 and a modified fgsea script was used to generate GSEA plots. Finally, the feature count matrix was also used to produce normalized TPM values for all genes in each sample; these were then plotted with ComplexHeatmap v2.6.2 (*Gu et al., 2016*). All scripts are deposited in GitHub (copy archived at *Smolkin, 2022*). For the analysis of GSE720181, the data set was downloaded as a featureCounts matrix and converted to TPM values in R v4.0.5.

## Metaphase spreads

Metaphase chromosome spreads were prepared by incubating cells with 100 ng/ml KaryoMAX Colcemid Solution in PBS (Gibco) for 3 hr. Cells were harvested at 1000 rpm and resuspended in 75 mM KCl at 37°C for 15 min. Cells were fixed in a 3:1 mixture of ice-cold methanol/acetic acid at least overnight at –20°C. Samples were then dropped onto pre-cleaned slides, briefly steamed (<5 s) over an 80°C water bath to disperse nuclei and air-dried overnight at room temperature. Slides were stained in Giemsa solution and mounted using Fisher Chemical Permount Mounting Medium (Thermo Fisher Scientific). Images were acquired on an Olympus IX50-S8F microscope using a 100× objective and images were analyzed using ImageJ.

## Telomere FISH

Metaphase chromosome spreads were prepared and dropped onto slides as described above. Instead of Giemsa staining, samples were treated with 100 µg/ml RNAse A for 1 hr at 37°C, dehydrated with a series of 70%, 90%, and 100% ethanol for 5 min each at room temperature, then allowed to air dry. Hybridization with 0.5 µg/ml CY-3 $(CCCTAA)_3$ probe (PNA Bio) was carried out in hybridization buffer (10 mM Tris pH 7.5, 70% formamide, 0.5% blocking reagent (Roche)). Samples were denatured at 75°C for 5 min and hybridization was then allowed to proceed at room temperature for 16 hr. Slides were washed two times in wash buffer (10 mM Tris pH 7.5, 0.1% BSA, and 70% formamide) and then three times in PBS/0.15% Triton X. Slides were subsequently incubated for 10 min at room temperature in SYTOX Green Nucleic Acid Stain (diluted to 0.5 mM in PBS). After a final PBS wash, slides were mounted with ProLong Gold Antifade Mountant with DAPI (Invitrogen). Images were acquired on a DeltaVision Elite Cell Imaging System (GE Healthcare Life Sciences) with a CMOS Camera on an Olympus IX-71 microscope using a 60× objective. Images were analyzed using ImageJ.

## Statistical analysis

Graphical representation of data and statistical analyses were performed using Prism 9 (GraphPad Software). Tables were prepared using Numbers 11 (Apple).

## Acknowledgements

The authors would like to thank the past and present members of the Chaudhuri and the Petrini labs for technical assistance, productive discussions, and constructive feedback. The authors specifically thank Youngjun Kim for experimental contributions. JC was supported by grants from the NIH (R01AI072194, R01AI124186, R56AI072194, U54CA137788, and P30CA008748), the Starr Cancer Research Foundation, the Ludwig Center for Cancer Immunotherapy, MSKCC Functional Genomics, and the Geoffrey Beene Cancer Center. JHJP was supported by grants from the NIH (R01GM56888, R35GM136278, U54OD020355, and P30CA008748). The authors thank Antonio Bravo for help with maintenance of the mouse colony. The authors acknowledge the use of the Memorial Sloan Kettering Cancer Mouse Genetics Core Facility.

## Additional information

### Funding

| Funder | Grant reference number | Author |
|---|---|---|
| National Institutes of Health | R01AI072194 | Jayanta Chaudhuri |
| National Institutes of Health | R01AI124186 | Jayanta Chaudhuri |
| National Institutes of Health | R56AI072194 | Jayanta Chaudhuri |
| National Institutes of Health | U54CA137788 | Jayanta Chaudhuri |
| National Institutes of Health | P30CA008748 | John HJ Petrini Jayanta Chaudhuri |
| National Institutes of Health | R01GM56888 | John HJ Petrini |
| National Institutes of Health | R35GM136278 | John HJ Petrini |
| National Institutes of Health | U54OD020355 | John HJ Petrini |
| Geoffrey Beene Cancer Research Center | | Jayanta Chaudhuri |

The funders had no role in study design, data collection and interpretation, or the decision to submit the work for publication.

### Author contributions

Keith Conrad Fernandez, Conceptualization, Formal analysis, Investigation, Visualization, Methodology, Writing – original draft, Writing – review and editing; Laura Feeney, Formal analysis, Investigation, Visualization, Methodology, Writing – original draft; Ryan M Smolkin, Software, Formal analysis, Writing – review and editing; Wei-Feng Yen, Allysia J Matthews, Formal analysis, Investigation, Methodology; William Alread, Investigation; John HJ Petrini, Jayanta Chaudhuri, Conceptualization, Resources, Supervision, Funding acquisition, Project administration, Writing – review and editing

### Author ORCIDs

Keith Conrad Fernandez ⓘ https://orcid.org/0000-0003-3757-8271
Jayanta Chaudhuri ⓘ https://orcid.org/0000-0002-3838-0075

### Ethics

All mice were housed and maintained in groups of five under specific pathogen-free conditions, and euthanized at the time of analyses in accordance with guidelines for animal care established by Memorial Sloan Kettering Cancer Center Research Animal Resource Center and the Institutional Animal Care and Use Committee (IACUC). All mouse experimentation protocols were approved by MSK's IACUC (Protocol Number: 05-12-030).

### Decision letter and Author response

Decision letter https://doi.org/10.7554/eLife.77073.sa1
Author response https://doi.org/10.7554/eLife.77073.sa2

## Additional files

### Supplementary files

Transparent reporting form

## Data availability

The RNAseq data and analysis generated in this study are deposited in GEO under the accession code GSE195734. The Gen1 and Mus81 expression data in the various B cell subsets was previously generated by Brazão et al. (2016) and deposited under GSE72018 in GEO.

The following dataset was generated:

| Author(s) | Year | Dataset title | Dataset URL | Database and Identifier |
|---|---|---|---|---|
| Fernandez K, Smolkin R, Chaudhuri J | 2022 | mRNA sequencing of control and resolvase-deficient ex vivo-stimulated B cells | https://www.ncbi.nlm.nih.gov/geo/query/acc.cgi?acc=GSE195734 | NCBI Gene Expression Omnibus, GSE195734 |

The following previously published dataset was used:

| Author(s) | Year | Dataset title | Dataset URL | Database and Identifier |
|---|---|---|---|---|
| Brazão TF, Johnson JS, Müller J, Heger A, Ponting CP, Tybulewicz VLJ | 2016 | Long non-coding RNAs in B cells (RNA-Seq) | https://www.ncbi.nlm.nih.gov/geo/query/acc.cgi?acc=GSE72018 | NCBI Gene Expression Omnibus, GSE72018 |

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
