## [Editor Report]

This manuscript is of interest to individuals working on genome stability and B lymphocyte development. Using knockouts for the genes encoding the structure-selective endonucleases GEN1 and MUS81 in mice, the authors show that the absence of both proteins is incompatible with embryonic development. On the background of a GEN1 knockout, a MUS81 flox allele was used to study the effect on B-cell development using the Mb1-Cre and Cd23-Cre drivers, showing that the absence of both proteins leads to development and maturation defects. Selective loss in mature B cells inhibited germinal center formation. This is the first study of these enzymes in an organismic context and in primary cells, revealing insight into the in vivo consequences of loss of GEN1 and MUS81 functions not previously accessible through studies in cultured cells.

---

## [Decision Letter]

**Decision letter after peer review:**

Thank you for submitting your article "The Structure-Selective Endonucleases GEN1 and MUS81 are Functionally Complementary in Safeguarding the Genome of Proliferating B Lymphocytes" for consideration by *eLife*. Your article has been reviewed by 3 peer reviewers, including Wolf-Dietrich Heyer as Reviewing Editor and Reviewer #1, and the evaluation has been overseen by Jessica Tyler as the Senior Editor.

Essential revisions:

1) The authors refer to GEN1 and MUS81 as Holliday junction resolvases. This terminology mischaracterizes the enzymes and inherently limits the type of DNA substrates that may be involved here. GEN1 can cleave HJs but also has a very broad substrate spectrum cleaving many types of branched and junction DNA molecules. MUS81 by itself is inactive, it requires EME1 to become a functional nuclease. MUS81-EME1 cannot cleave HJs, only in the context of the tri-nuclease *SLX1*-SLX4, MUS81-EME1, XPF-ERCC1. While I suspect the authors use this terminology as a shorthand and are aware of these complexities, many readers are not and will be misled. For this reason, the terminology, description, and discussion of these enzymes should be more precise. Both enzymes are structure-selective endonucleases. Just focusing on the HJ intermediate is also too narrow, as SDSA does not involve HJs and may be relevant here. MUS81-EME1 can cleave D-loops, another central HR intermediate, which may be of significance here. For these reasons, the concluding sentence of the introduction (line 135 ff) and many other passages should be rephrased to broaden the discussion of which specific molecular defects (dHJ resolution, other junction substrates, including stalled replication forks, etc.) underlie the observed phenotypes.

Note in Line 106, also MUS81-EME1 can cleave replication fork-type substrates.

2) The Mus81 flox allele should be described in more detail to indicate what exactly is predicted to be deleted in the encoded protein.

3) The results from the cell cycle analysis underline the indispensable role of GEN1 and MUS81 in supporting proliferation of activated B lymphocytes. These conclusions disregard another potentially interesting observation. The peak of the double knock-out splenocytes accumulated at 0 cell divisions (Figure 3B) is interpreted as cells that have not divided. How does deletion of proteins that participate in late homologous recombination repair affect the initiation of cell division? Could the authors provide a possible explanation?

4) Figure 5: The data presented on DNA damage are not convincing. It is the primary evidence that defective HR resolution is the cause, i.e. breaks at the same place on both sister arms. The images provided with damage classifications can be interpreted to either be different types of damage or undamaged chromosomes:

a) The picture labeled exchange can be caused by metaphase dropping – chromosome arms lay over each other does not mean that crossover occurred. It appears to be a radial or radial-like chromosome fusion.

b) The image labeled acentric is not an acentric chromosome, rather it is 2 chromosomes with 2 centromeres – the twin telomere foci in the middle look like the acrocentric arms at the centromere. Again, it is likely a fusion/damage event but appears to have a centromere by DNA + telomere staining.

c) The chromosome labeled dicentric has only one centromere at the top – again the twin telomere dots label the acrocentric arms at the centromere.

A commercially available centromere probe would aid with chromosome rearrangement analyses.

The interpretation in the manuscript relies heavily on published results to support the claim that the DKO mice are suffering from defective double Holliday junction (dHJ) resolution. However, if a dHJ persisted into mitosis without being resolved, why would this manifest as a chromosome break in a metaphase chromosome spread where the spindle has been destabilized with colcemid to "trap" cells in mitosis? Rather chromosome arms should be intertwined at the dHJ site which may not be visible at all, or visible as a "pinch" in the chromosome (picture 5F top left may actually be this, the "pinch" looking like a break -the DNA may be decondensed because of a HJ/dHJ?). This may be visible in the spreads: there are 4 chromosomes with this gap in them on both arms in Figure 5A. The HJ/dHJ and resulting UFB would only snap and create a "true" DSB on both sisters in anaphase when the spindle pulls the two sisters apart.

An increase in chromosome breaks is also a hallmark of defective NHEJ – double-strand breaks occurring in G1 are not repaired and then replicated leading to a chromosome break at same places on the sister. While this is unlikely for Gen1/Mus81-deficient cells, it is still formally possible.

5) Western Blot or genomic DNA analysis are necessary to assess the efficiency of Cre-mediated gene deletion in the different bone marrow cell populations. Deletion may be incomplete due to the late stage at which it acts and explain why germinal centers cells are only reduced ~2-fold in Figure 2A? The discussion regarding germinal center B cells independent of GEN1-MUS81 expression could be expanded.

6) The discussion of the results of the RNA-Seq analysis should be expanded. One of the most significantly downregulated pathways in the RNA-Seq are related to MYC targets. How does the deletion of the two resolvases connect to MYC?

7) The results of the RNA-Seq study are poorly discussed. While there is a clear experimental follow up on p53 pathway and apoptosis activation, What are the possible implications or causes of the interferon α response upregulation?

*Reviewer #1 (Comments for the Authors):*

Essential revision

1) The authors refer to GEN1 and MUS81 as Holliday junction resolvases. This terminology mischaracterizes the enzymes and inherently limits the type of DNA substrates that may be involved here. GEN1 can cleave HJs but also has a very broad substrate spectrum cleaving many types of branched and junction DNA molecules. MUS81 by itself is inactive, it requires EME1 to become a functional nuclease. MUS81-EME1 cannot cleave HJs, only in the context of the tri-nuclease *SLX1*-SLX4, MUS81-EME1, XPF-ERCC1. While I suspect the authors use this terminology as a shorthand and are aware of these complexities, many readers are not and will be misled. For this reason, the terminology, description, and discussion of these enzymes should be more precise. Both enzymes are structure-selective endonucleases. Just focusing on the HJ intermediate is also too narrow, as SDSA does not involve HJs and may be relevant here. MUS81-EME1 can cleave D-loops, another central HR intermediate, which may be of significance here. For these reasons, the concluding sentence of the introduction (line 135 ff) should be rephrased.

Note in Line 106, MUS81-EME1 can also cleave replication fork-type substrates.

2) The Mus81 flox allele should be described in more detail to indicate what exactly is predicted to be deleted in the encoded protein.

*Reviewer #2 (Comments for the Authors):*

1) The results from the cell cycle analysis underline the indispensable role of GEN1 and MUS81 in supporting proliferation of activated B lymphocytes. These conclusions disregard another potentially interesting observation. The peak of the double knock-out splenocytes accumulated at 0 cell divisions (Figure 3B) is interpreted as cells that have not divided. How does deletion of proteins that participate in late homologous recombination repair affect the initiation of cell division? Could the authors provide a possible explanation?

2) Since the authors assign the cell cycle block and increased apoptosis of the activated GEN1-MUS81 deficient B cells to p53-dependent pathways (Figure 4C), would a p53-deficient background rescue the phenotype observed in Figure 3? In reference to point 1 above, it would be of interest to see the consequences of p53 deficiency on the accumulation of the undivided population.

3) On the same line, one of the most significantly downregulated pathways in the RNA-Seq are related to MYC targets. How does the deletion of the two resolvases connect to MYC?

4) The results of the RNA-Seq study are poorly discussed. While there is a clear experimental follow up on p53 pathway and apoptosis activation, the authors do not discuss the possible implications or causes of the interferon α response upregulation.

*Reviewer #3 (Comments for the Authors):*

1) Adding class switch recombination frequencies for WT, Gen1, Mus81 and double-deficient cells would be appropriate to determine what programmed damage is occurring in this cell population. Though it is likely reduced due to low proliferation, it would still be interesting to see.

2) The data presented on DNA damage is not convincing. The images provided with damage classifications can be interpreted to either be different types of damage or undamaged chromosomes:

a) the picture labeled exchange can be caused by metaphase dropping – chromosome arms lay over each other does not mean that crossover occurred. It appears to be a radial or radial-like chromosome fusion.

b) The image labeled acentric is not an acentric chromosome, rather it is 2 chromosomes with 2 centromeres – the twin telomere foci in the middle look like the acrocentric arms at the centromere. Again it is likely a fusion/damage event but appears to have a centromere by DNA + telomere staining.

c) The chromosome labeled dicentric has only one centromere at the top – again the twin telomere dots label the acrocentric arms at the centromere.

A commercially available centromere probe would aid with chromosome rearrangement analyses.

3) Additional experiments to connect the developmental phenotypes to a defect in recombinational repair or replication stress would strengthen the conclusions of the manuscript in its current form. Do these cells exhibit reduced replication fork progression by DNA fiber analysis and/or activation of replication stress checkpoints (ATR / Chk1 phosphorylation)?

---

## [Author Response]

Essential revisions:1) The authors refer to GEN1 and MUS81 as Holliday junction resolvases. This terminology mischaracterizes the enzymes and inherently limits the type of DNA substrates that may be involved here. GEN1 can cleave HJs but also has a very broad substrate spectrum cleaving many types of branched and junction DNA molecules. MUS81 by itself is inactive, it requires EME1 to become a functional nuclease. MUS81-EME1 cannot cleave HJs, only in the context of the tri-nuclease SLX1-SLX4, MUS81-EME1, XPF-ERCC1. While I suspect the authors use this terminology as a shorthand and are aware of these complexities, many readers are not and will be misled. For this reason, the terminology, description, and discussion of these enzymes should be more precise. Both enzymes are structure-selective endonucleases. Just focusing on the HJ intermediate is also too narrow, as SDSA does not involve HJs and may be relevant here. MUS81-EME1 can cleave D-loops, another central HR intermediate, which may be of significance here. For these reasons, the concluding sentence of the introduction (line 135 ff) and many other passages should be rephrased to broaden the discussion of which specific molecular defects (dHJ resolution, other junction substrates, including stalled replication forks, etc.) underlie the observed phenotypes.Note in Line 106, also MUS81-EME1 can cleave replication fork-type substrates.

We would like to thank the reviewers for bringing this up and in response, we have changed the terminology and description of these enzymes, referring them as ‘structure-selective endonucleases’ consistently throughout the manuscript to reflect their broad substrate specificity and to minimize reader confusion. We also have modified and expanded the Introduction and Discussion sections to recognize and elaborate on the possibility that the molecular defects underpinning the developmental and immunological phenotypes observed in the B-cell-specific *Gen1*-*Mus81*-null mice might involve the persistence of other non-HJ joint and branched structures that necessitate GEN1 and MUS81 processing, consequently impairing a greater variety of DNA replication and recombination events beyond what we have described initially.

2) The Mus81 flox allele should be described in more detail to indicate what exactly is predicted to be deleted in the encoded protein.

We have revised the text to describe the location of the loxP sites and the region of the protein that is being deleted in the floxed *Mus81* mouse.

3) The results from the cell cycle analysis underline the indispensable role of GEN1 and MUS81 in supporting proliferation of activated B lymphocytes. These conclusions disregard another potentially interesting observation. The peak of the double knock-out splenocytes accumulated at 0 cell divisions (Figure 3B) is interpreted as cells that have not divided. How does deletion of proteins that participate in late homologous recombination repair affect the initiation of cell division? Could the authors provide a possible explanation?

This is a very astute comment by the reviewer. In response, we quantified the absolute number of division- zero cells under all conditions of stimulation and found it to similar across all genotype groups, indicating that a small subset of B cells does not undergo cell division under any condition. As fewer viable, proliferating DKO cells were present in the 72-hour culture, the undivided (division-zero) cells are thus disproportionately represented in the DKO B cell culture compared with those in the control and SKO B cell cultures, producing the “enriched” division-zero peak in Figure 3C. We have included this data in Figure 3D and Figure 3—figure supplement 1C.

And, as the reviewer has noted, while GEN1 and MUS81 have canonically been associated with the resolution of structural intermediates generated during HR (primarily HJs), they could also act on a more diverse array of DNA substrates, including those formed independently of recombinational DSB repair (i.e., during replication). Therefore, the inability of DKO cells to initiate cell division or sustain multiple rounds of replication could reflect a failure to eliminate replication-induced junction molecules, the persistence of which leads to G2/M stalling and abrogation of a complete cell division. These possibilities have now been described in the revised manuscript.

4) Figure 5: The data presented on DNA damage are not convincing. It is the primary evidence that defective HR resolution is the cause, i.e. breaks at the same place on both sister arms. The images provided with damage classifications can be interpreted to either be different types of damage or undamaged chromosomes:a) The picture labeled exchange can be caused by metaphase dropping – chromosome arms lay over each other does not mean that crossover occurred. It appears to be a radial or radial-like chromosome fusion.

We thank the reviewer for this comment. We agree with the reviewer, and we have corrected the labeling to read as “radial”. We are confident that we have counted radial structures in our analysis, and we have included additional examples of what has been identified as a radial in Figure 5—figure supplement 1.

b) The image labeled acentric is not an acentric chromosome, rather it is 2 chromosomes with 2 centromeres – the twin telomere foci in the middle look like the acrocentric arms at the centromere. Again, it is likely a fusion/damage event but appears to have a centromere by DNA + telomere staining.

We have called these structures “acentrics” because they appear to be so by Giemsa staining; only by Tel-FISH could one observe a radial structure involving the short arms of two chromosomes. We agree that this should have been addressed in the original text, and we have added discussion to this effect and additional examples of 'acentrics' in Figure 5—figure supplement 1.

c) The chromosome labeled dicentric has only one centromere at the top – again the twin telomere dots label the acrocentric arms at the centromere.A commercially available centromere probe would aid with chromosome rearrangement analyses.

This chromosome appears to have two centromeres due to the additional constriction, but it is impossible to be certain of this without doing centromere staining (although mouse chromosomes are acrocentric, lack of additional telomere signal does not preclude an additional centromere being present). This is one instance where we respectfully disagree with the reviewer. We believe that performing the centromere staining would not add significant value to the paper given that dicentrics do not represent a major class of aberration in the DKO cells and removing the dicentrics from our analysis would not impact the conclusions of the paper.

The interpretation in the manuscript relies heavily on published results to support the claim that the DKO mice are suffering from defective double Holliday junction (dHJ) resolution. However, if a dHJ persisted into mitosis without being resolved, why would this manifest as a chromosome break in a metaphase chromosome spread where the spindle has been destabilized with colcemid to "trap" cells in mitosis? Rather chromosome arms should be intertwined at the dHJ site which may not be visible at all, or visible as a "pinch" in the chromosome (picture 5F top left may actually be this, the "pinch" looking like a break -the DNA may be decondensed because of a HJ/dHJ?). This may be visible in the spreads: there are 4 chromosomes with this gap in them on both arms in Figure 5A. The HJ/dHJ and resulting UFB would only snap and create a "true" DSB on both sisters in anaphase when the spindle pulls the two sisters apart.

We agree that the breaks could be persistent HJs disrupting chromosome condensation, as has been previously suggested in the literature. However, prior to the chromosome analysis, some of these cells would have been through multiple cell cycles, so these could also be true breaks that are the result of an unresolved HJ that was broken in the previous mitosis. Although not directly tested by us in our DKO cells, the West lab recently showed that these unresolved recombination intermediates form ultrafine anaphase bridges that are broken during cytokinesis, generating DNA ends amenable to NHEJ-mediated chromosomal fusions (Chan et al., 2018, *Nature Cell Biology*). We have discussed these possibilities in the revised manuscript.

An increase in chromosome breaks is also a hallmark of defective NHEJ – double-strand breaks occurring in G1 are not repaired and then replicated leading to a chromosome break at same places on the sister. While this is unlikely for Gen1/Mus81-deficient cells, it is still formally possible.

We agree with the reviewer, and we have now acknowledged this possibility in the revised manuscript.

5) Western Blot or genomic DNA analysis are necessary to assess the efficiency of Cre-mediated gene deletion in the different bone marrow cell populations. Deletion may be incomplete due to the late stage at which it acts and explain why germinal centers cells are only reduced ~2-fold in Figure 2A? The discussion regarding germinal center B cells independent of GEN1-MUS81 expression could be expanded.

We performed quantitative genomic PCR to determine the efficiency of *Mus81* deletion, and RT-qPCR to measure *Gen1* and *Mus81* mRNA expressions in Fractions B+C (pro-B and pre-B) cells of *Gen1*^+/–^
*Mus81*^fl/fl^, *Gen1*^–/–^
*Mus81*^fl/fl^, and *Mb1*-Cre: *Gen1*^+/–^
*Mus81*^fl/fl^. *Mb1*-Cre: *Gen1*^+/–^
*Mus81*^fl/fl^ mice do not possess Fractions B+C cells and so these analyses could not be performed. We have included the data in Figure 1—figure supplement 1E and 1F. We want to note that despite several attempts with various commercial and lab-made antibodies using protein amounts as high as 100 μg, we were unable to reliably detect mouse GEN1 and MUS81 proteins in our ex vivo B cell cultures.

We did not analyze the deletion efficiency in Fraction A (pre-pro-B) cells because we recently realized from unrelated experiments that the “Fraction A” cells defined by the gating strategy employed in our manuscript is composed predominantly (~90%) of plasmacytoid dendritic cells that are positive for B220 and negative for the same set of B-lineage markers used in our experiment. We were only able to identify these contaminating plasmacytoid dendritic cells by including anti-Siglec-H and anti-Cd11c antibodies in our staining panel. (We now realize this is a common omission made by others in the field.) We have therefore decided to remove all data related to the Fraction A population from Figure 1 and have revised the Results section. The conclusion that the loss of *Gen1* and *Mus81* in early B-cell progenitors impairs the development of mature B cells, however, is unaffected despite the exclusion of the data as Fraction B cells are almost completely absent in the *Mb1-Cre*: *Gen1*^+/–^
*Mus81*^fl/fl^ mice.

6) The discussion of the results of the RNA-Seq analysis should be expanded. One of the most significantly downregulated pathways in the RNA-Seq are related to MYC targets. How does the deletion of the two resolvases connect to MYC?

In agreement with this comment, we have expanded the discussion of the RNA-seq findings. Porter et al., (2017, *Molecular Cell*) reported that p53 upregulation in response to DNA damage suppresses MYC transcription via the binding of p53 to an enhancer region within a MYC super-enhancer element, and that such transcriptional inhibition is necessary for the proper enforcement of p53-dependent cell cycle arrest in cells experiencing a high level of genotoxic stress. This MYC-counteracting mechanism would thus be similarly engaged in DKO cells that exhibit gross genomic destabilization and high p53 activity. Furthermore, the transcriptional targets of p53 can directly suppress MYC expression via other transcriptional (by the lncRNA *Pvt1b*) and post-transcriptional (by miR-145 and miR-34b) mechanisms, and indirectly via the inhibition of mTOR (by TSC2 and PTEN), the upstream regulator of MYC (Cui et al., 2021, *Frontiers in Cell and Developmental Biology*; Levine, 2020, *Nature Reviews Cancer*; Olivero et al., 2020, *Molecular Cell*; Sachdeva et al., 2009, *PNAS*).

In addition, we have generated and extensively characterized the *Cd23*-Cre: *Gen1*^-/-^
*Mus81*^fl/fl^
*Trp53*^fl/fl^ mice to examine whether the inactivation of p53 can ameliorate the DKO phenotype. Loss of p53 did not improve any of the defects observed in the DKO cells, suggesting that the DNA damage sustained by the proliferating DKO cells might be too extensive for their survival. These findings are now presented in the Results section and included in Figure 4—figure supplement 2.

7) The results of the RNA-Seq study are poorly discussed. While there is a clear experimental follow up on p53 pathway and apoptosis activation, What are the possible implications or causes of the interferon α response upregulation?

We apologize for the brevity of the RNA-seq discussion. In the revised text, we have now expanded on how the sensing of cytosolic DNAs (e.g., micronuclei, free DNAs) by cGAS and the subsequent activation of an immune signaling cascade induces the type I IFN transcriptional program. We have also elaborated on the role type I IFN signaling might play in DNA-stimulated cell death, particularly in activating effector proteins involved in apoptosis and necroptosis (Paludan et al., 2019, *Nature Reviews Immunology*). Further, activation of the adaptor protein STING by cGAS has recently been implicated in the upregulation of p53 signaling in T cells, and this could potentially occur in our ex vivo-activated endonuclease-deficient B cells to elicit apoptosis (Concepcion et al., 2022, *Nature Immunology*).